# Unsupervised Feature Selection towards Maximizing Pattern Discrimination Power

**Wangduk Seo**[1]                                         **Jaesung Lee**[1,2*]

[1]AI/ML Innovation Research Center, Chung-Ang University, Seoul, South Korea
[2]Department of Artificial Intelligence, Chung-Ang University, Seoul, South Korea

## Abstract

The goal of unsupervised feature selection is to identify a feature subset based on the intrinsic characteristics of a given dataset without user-guided information such as class variables. To achieve this, score functions based on information measures can be used to identify essential features. The major research direction of conventional information-theoretic unsupervised feature selection is to minimize the entropy of the final feature subset. Although the opposite way, i.e., maximization of the joint entropy, can also lead to novel insights, studies in this direction are rare. For example, in the field of information retrieval, selected features that maximize the joint entropy of a feature subset can be effective discriminators for reaching the target tuple in the database. Thus, in this work, we first demonstrate how two feature subsets, each obtained by minimizing/maximizing the joint entropy, respectively, are different based on a toy dataset. By comparing these two feature subsets, we show that the maximization of the joint entropy enhances the pattern discrimination power of the feature subset. Then, we derive a score function by remedying joint entropy calculation; high-dimensional joint entropy calculation is circumvented by using the low-order approximation. The experimental results on 30 public datasets indicate that the proposed method yields superior performance in terms of pattern discrimination power-related measures.

## 1 INTRODUCTION

Recent advancements in storage technology have led to exponential growth of data, such as the web ecosystem [Brickley et al., 2019]. This proliferation of data poses significant challenges in distinguishing meaningful patterns within the vast complexity and volume of information [Puerto-Santana et al., 2023]. In particular, when the number of features becomes excessively large, patterns in the dataset can lose their discriminative power since the similarity among all patterns becomes similar, making subsequent analysis unreliable [Watanabe, 1969]. In such contexts, Unsupervised Feature Selection (UFS) emerges as an effective approach, offering a method to select core features from the original dataset.

The goal of UFS is to reduce the number of features needed for data representation while maintaining the essential information [Li et al., 2012, Shang et al., 2022, Karami et al., 2023, Feng et al., 2016, Wang et al., 2015]. Because there is no user-guided information, such as class variables, UFS methods should identify a feature subset based on the intrinsic characteristics of the dataset. By eliminating unnecessary features, the high dimensionality of the dataset can be remedied, and hence novel patterns [Wang et al., 2023] in the dataset can easily be identified. Regarding this, information entropy is known as a popular tool for measuring the information content of a variable set.

Conventional studies of information-theoretic UFS predominantly focus on measuring the relevancy or redundancy of features in the original dataset and selecting essential features [Hu et al., 2022, Wang et al., 2022]. In this framework, the algorithms are often designed to minimize the entropy of feature pairs in candidate feature subsets[1], which is a major research direction of current UFS studies [Zhu et al., 2023, Zhang et al., 2023]. By comparison, the study of maximizing the entropy of feature subsets is pretty rare, even though this strategy can also lead to novel insights where the discrimination power of patterns is important. To achieve this, we can consider a UFS process that maximizes the information content or the pattern discrimination power

---

*Corresponding author

[1]It should be noted that, in the information theory, the minimization of joint entropy of feature pairs is equivalent to the maximization of mutual information of feature pairs.

*Accepted for the 40[th] Conference on Uncertainty in Artificial Intelligence* (UAI 2024).

of selected feature subset for the given dataset, which can be viewed as the opposite direction of conventional studies.

In this work, we propose an information-theoretic UFS method that identifies a compact yet effective feature subset by maximizing entropy. First, we will demonstrate how two feature subsets, each obtained by minimizing/maximizing the joint entropy of features to be selected, respectively, are different based on a simple toy dataset. Our example shows that the maximization of joint entropy leads to the enhancement of the pattern discrimination power for given dataset; all the patterns in the datasets can be discriminable with a minimal number of features. Next, we derive our score function for information-theoretic UFS based on the joint entropy maximization. Unfortunately, it is well-known that the estimation of high-dimensional joint entropy is demanding in practice due to a limited number of patterns [Seo et al., 2019, Lee and Kim, 2013]. As a possible solution, we remedy the computation of high-dimensional joint entropy from a large number of features by decomposing it into the sum of low-dimensional joint entropy terms instead of introducing heuristic approaches [Yuan et al., 2021b,a]. Finally, we validated the performance of the proposed UFS method using 30 public datasets and confirmed its superiority in terms of pattern discrimination power-related measures. The main contributions of this work are summarized as follows:

- An information-theoretic UFS method is introduced, which maximizes entropy to identify an effective feature subset, thereby significantly enhancing pattern discrimination power within datasets.

- A comparative analysis demonstrates our approach using a toy dataset, illustrating the differences between feature subsets obtained through entropy minimization and maximization, and highlighting the enhanced pattern discrimination power achieved with entropy maximization.

- To tackle the computational challenge of high-dimensional joint entropy, we introduce a novel score function for UFS based on joint entropy decomposition.

- The efficacy of the proposed method is validated through extensive testing on 30 public datasets, which confirms its superiority in improving pattern discrimination power over existing UFS methods.

## 2 RELATED WORK

The primary aim of UFS is to reduce the dimensionality of data while preserving the inherent structure useful for subsequent tasks. These UFS methods can be roughly categorized according to their strategy for FS as filter, wrapper, and hybrid methods [Solorio-Fernández et al., 2020]. Among them, the filter methods that do not assume a fixed learning algorithm can be further divided into two groups: univariate and multivariate approaches. Typically, the multivariate unsupervised feature filters yield better performance than the univariate feature filters because the multivariate approach is able to consider the relation among features to be selected. In this work, we focus on the multivariate feature filter approach.

A majority of UFS methods are concentrated on preserving local structures and identifying latent labels of the given dataset, relying on similarity among data patterns. In this regard, He et al. [2006] introduced the Laplacian score, a method that ranks features based on how well they preserve the local structure of data. Similarly, Cai et al. [2010] proposed a multi-cluster FS technique designed to maintain the multi-cluster characteristics of the data. Other works tried to incorporate additional information for better identification of local structures. For instance, Yang et al. [2011] suggested an unsupervised discriminative FS method that integrates both discriminative information and intrinsic data structure using an $l_{2,1}$-norm. Li et al. [2012] devised a UFS algorithm with a non-negative constraint, utilizing non-negative matrix factorization to construct the projection matrix for FS. Zhu et al. [2023] leveraged an $l_{2,0}$-norm constraint to perform group UFS. Lastly, Villa et al. [2021] proposed a radial basis kernel (U2) to manage non-linearity within the conventional UFS framework of $l_{2,1}$-norm regularization.

In another group of studies, the UFS was embedded into a subsequent unsupervised learner, such as the clustering algorithm. Wang et al. [2015] combined clustering algorithms with a UFS process in their embedded UFS method. Miao et al. [2022] also presented an embedded technique based on graph regularization, which is capable of preserving the local reconstruction relationships among neighboring data points. Shang et al. [2022] proposed a UFS approach that utilizes non-negative spectral feature learning with an adaptive rank constraint. This adaptive constraint enables the algorithm to update the local structure more accurately during the UFS process. Zhang et al. [2020] incorporated an adaptive graph learning constraint to integrate a similarity matrix into the existing UFS framework. Recently, Karami et al. [2023] introduced a variance-covariance distance (VCSD) to tackle both dimensionality reduction and subspace learning.

Numerous UFS methods have been developed to account for the information within the data. Faivishevsky and Goldberger [2012] introduced a UFS method that estimates mutual information (MI) between features. Instead of relying on a parametric model for this calculation, their approach uses statistical dependencies between features. Yuan et al. [2021a] developed a UFS method grounded in fuzzy rough set theory for handling mixed data. Specifically, their method constructs a fuzzy information system from the original data and selects features based on a fuzzy dependency function, thereby maximizing feature relevance. Another notable extension is Fuzzy MI (FMI), which reformulates

Table 1: A toy dataset composed of six patterns and four categorical features ($f_1$, $f_2$, $f_3$, and $f_4$), and two selected feature subsets guided by minimizing ($min$) and maximizing ($max$) joint entropy, respectively.

| Pattern | Original Features | | | | Entropy-based UFS | | | |
| | | | | | $min$ | | $max$ | |
| | $f_1$ | $f_2$ | $f_3$ | $f_4$ | $f_1$ | $f_2$ | $f_3$ | $f_4$ |
| --- | --- | --- | --- | --- | --- | --- | --- | --- |
| $p_1$ | A | A | A | A | A | A | A | A |
| $p_2$ | B | A | B | A | B | A | B | A |
| $p_3$ | A | B | C | A | A | B | C | A |
| $p_4$ | A | B | A | B | A | B | A | B |
| $p_5$ | A | B | B | B | A | B | B | B |
| $p_6$ | A | B | C | B | A | B | C | B |

MI under the fuzzy theory [Yuan et al., 2021b]. However, these methods often entail a high computational cost for calculating MI between features and additionally require the tuning of hyperparameters. Feng et al. [2016] proposed an efficient UFS method for hyperspectral image datasets by employing heuristic high-dimensional MI estimation. Similarly, Lim and Kim [2021] incorporated pairwise MI into the spectral learning framework. A common drawback of those approaches is that the analysis of the final feature subset may not be theoretically supported because the score function is devised heuristically. In addition, they are unsuitable for maximizing the pattern discrimination power of data, which is the primary focus of this work.

# 3  PROPOSED METHOD

## 3.1  MOTIVATION

Conventional information-theoretic UFS methods often concentrate on selecting features that decrease the dissimilarity among patterns, thereby enhancing the performance of subsequent learners, such as the clustering algorithm. In this context, the objective function can be designed to minimize the joint entropy of the final feature subset $S$, which can be represented as $\arg\min_S H(S)$ where $H(S) = -\sum P(S) \log P(S)$. In contrast, we may search for a feature subset by optimizing $\arg\max_S H(S)$.

To demonstrate how the two objective functions lead to different final feature subsets, we created a toy dataset consisting of six patterns and four categorical features ($f_1$, $f_2$, $f_3$, and $f_4$) as shown in Table 1. The toy dataset shows that none of a single feature can distinguish all the patterns; for example, patterns $p_1, p_3, p_4, p_5$ and $p_6$ are indiscriminable in the viewpoint of $f_1$ because they are assigned to the same category, $A$. Suppose that we want to identify the optimal feature subset where $|S| = 2$ that minimizes $H(S)$. This problem can be solved by instantiating all the feature pairs, measuring the joint entropy values, and then choosing a

feature subset that yields a minimal $H(S)$ value.

The second rightmost column of Table 1, namely, $min$, shows the final feature subset of this case where $f_1$ and $f_2$ are selected, with a joint entropy of 1.252. In this case, there are three groups $\{p_1\} = (A, A)$, $\{p_2\} = (B, A)$, and $\{p_3, p_4, p_5, p_6\} = (A, B)$ that are discriminable to each other. In other words, patterns $p_3, p_4, p_5$, and $p_6$ are indiscriminable to each other. Next, the rightmost column $max$ of Table 1 shows the final feature subset $S = \{f_3, f_4\}$ obtained by maximizing $H(S)$, with a joint entropy of 2.585. In this case, there are six discriminable groups $\{p_1\} = (A, A)$, $\{p_2\} = (B, A)$, $\{p_3\} = (C, A)$, $\{p_4\} = (A, B)$, $\{p_5\} = (B, B)$, and $\{p_6\} = (C, B)$.

Our example shows that the strategy of maximizing $H(S)$ will lead to the selection of features that make discriminable groups as much as possible until all the patterns in the dataset become discriminable, i.e., a maximum pattern discrimination power is reached. The concept of maximizing $H(S)$ can be used to build an effective taxonomy of a system because it can reduce the number of discriminators in the system, and hence accelerate the retrieval time.

## 3.2  SCORE FUNCTION

Let $W \in \mathcal{R}^{|F|}$ be the original dataset with $|F|$ features $F = \{f_1, f_2, \cdots, f_{|F|}\}$ and the goal of the UFS is to identify a feature subset $S$ consisting of $n$ features with the optimal pattern discrimination power where $n$ is the number of features to be selected. Because there are $2^{|F|}$ possible feature subsets, it is impractical to identify the optimal feature subset by searching all possible feature subsets, i.e., the exhaustive search. To achieve this, UFS methods often employed an incremental search strategy to effectively instantiate candidate feature subsets. The incremental search starts with an empty set and iteratively adds a new feature to the subset of features until $|S|$ reaches $n$.

Owing to the monotonicity of entropy, the original feature set $F$ should have the largest entropy $H(F)$ or pattern discrimination power. However, because only $n \ll |F|$ features can be included in $S$, the original pattern discrimination power will be damaged after the FS process. To maintain the pattern discrimination power as much as possible, the difference between $H(F)$ and $H(S)$ should be minimized. Thus, the objective function can be written as

$$\arg\min_S (H(F) - H(S)). \quad (1)$$

Because $H(F)$ is constant, Equation (1) can be rewritten as

$$\arg\min_S \{H(F) - H(S)\} \propto \arg\min_S -H(S)$$
$$= \arg\max_S H(S). \quad (2)$$

Based on the incremental search strategy, the algorithm

must identify a new feature $f^+$ from $F \setminus S$ that maximally increases the entropy of feature subset $S$. Thus, the objective function $J$ can be represented as

$$J = \arg\max_{f^+ \in F \setminus S} H(S, f^+). \tag{3}$$

Because $H(S, f^+)$ can be a high-dimensional joint entropy term due to $S$, an accurate estimation may not be achieved in practice because of an insufficient number of patterns. To circumvent this issue, in this work, we estimate $H(S, f^+)$ by using low-order joint entropy terms involving only $k \ll |S|$ features, which is a frequently-used strategy in the field of information-theoretic FS. For brevity, we define $k$-cardinality entropy [Lee and Kim, 2015] as Definition 1.

**Definition 1.** *Sum of the $k$-cardinality entropy.*

$$U_k(X) = \sum_{Y \in X_k'} H(Y), \tag{4}$$

where $X'$ is the power set of $X$ and $X_k' = \{e | e \in X', |e| = k\}$.

Based on the Definition 1, the upper bound of Han's inequality [Han, 1978] can be rewritten as Proposition 1.

**Proposition 1.** *$k$-cardinality representation of Han's inequality*

$$H(X) \leq \frac{1}{n-1} U_{n-1}(X'), \tag{5}$$

where $n$ is the number of variables in $X$. Based on the Proposition 1, we get Lemma 1 by applying the upper bound to its subsequent joint entropy terms.

**Lemma 1.** *Let $U_k(S')$ be the $k$-cardinality entropy of given variable sets $S$. Then the lower bound and the upper bound of $U_k(S')$ can be defined as*

$$\frac{1}{k-1}\left(kU_k(S') - \binom{n-1}{k-1}U_1(S')\right) \leq U_k(S')$$
$$\leq \left(\frac{n-k+1}{k-1}\right)U_{k-1}(S'). \tag{6}$$

*Proof.* The detailed proof is provided in the work of Lee and Kim [2015]. □

Lemma 1 indicates that the upper bound of $U_k(S')$ is determined by the $(k-1)$-cardinality entropy term. Thus, by recursively applying Lemma 1, we can obtain the $k$-cardinality approximation of the high-dimensional joint entropy $H(X)$, as stated in Theorem 1.

**Theorem 1.** *Upper bound of the $H(X)$ with $k$-cardinality entropy is*

$$H(X) \leq \left(\prod_{i=1}^{b} \frac{i}{n-i}\right) U_k(X'), \tag{7}$$

*where $b = \min(n-k, k-1)$.*

*Proof.* The detailed proof is provided in the work of Seo et al. [2019]. □

Theorem 1 indicates that the upper bound becomes tighter when $k$ in Equation (7) is set to a large value, and hence a better estimation of $H(X)$ can be obtained. From the Theorem 1, $H(S, f^+)$ can be approximated based on the sum of the $k$-cardinality entropy if the upper bound is accepted as the estimator. In our experiments, we set $k$ to two because it is the minimum value for the score function being a multivariate feature filter[2], which is the main focus of this work. As a result, the objective function $J$ is approximated as

$$J \approx \arg\max_{f^+} \sum_{i=1}^{b} \frac{i}{|S|+1-i} U_2(\{S', f^+\}')$$
$$= \arg\max_{f^+} \frac{1}{|S|} U_2(\{S', f^+\}'), \tag{8}$$

where $b = min(|S| + 1 - 2, 2 - 1) = 1$. Because elements of the power set $\{S, f^+\}'$ can be divided into two parts, whether the element contains $f^+$ or not, Equation (8) can be rewritten as

$$J \approx \arg\max_{f^+} \frac{1}{|S|}\left(U_2(S') + U_2(f^+ \times S')\right), \tag{9}$$

where $\times$ denotes the Cartesian product between two sets. Because the $U_2(S')$ is constant to $f^+$, Equation (9) can be rewritten as

$$J \approx \arg\max_{f^+} U_2(f^+ \times S'). \tag{10}$$

Finally, $J$ can be approximated as

$$J \approx \arg\max_{f^+} \sum_{f \in S} H(f^+, f). \tag{11}$$

In the case of the $S = \{\emptyset\}$ where none of the features are selected yet, Equation (11) can be represented as

$$J \approx \arg\max_{f^+} H(f^+). \tag{12}$$

It is worth noting that Equation (12) is the $k$-cardinality entropy where $k$ is 1. However, the algorithm may start with the optimal feature pairs (Please refer to Section 4.2.)

---

[2]If $k$ is set to one, then a score function for univariate feature filter is instantiated because the maximum number of features to be considered is one when $k = 1$, thereby the relation among features, for example, feature pairs, cannot be considered.

**Algorithm 1** Incremental Search for the Proposed Method

1: $f^+ \leftarrow \arg\max_{f^+ \in F} H(f^+)$
2: $S \leftarrow \{f^+\}$
3: **while** $|S| < n$ **do**
4:     $f^+ \leftarrow \arg\max_{f^+ \in F-S} \sum_{f \in S} H(f^+, f)$
5:     $S \leftarrow S \cup \{f^+\}$
6: **end while**

### 3.3 INCREMENTAL SEARCH

The proposed method is designed as a model-free, non-parametric measure that only requires calculating information-theoretic quantities based on joint entropy calculation. Specifically, the proposed method incrementally selects a new feature $f^+$ from the $F \setminus S$ and adds it to the subset of features $S$. The Algorithm 1 depicts the incremental search process of the proposed method. First, $f^+$ is selected by Equation (12) and the $S$ is initialized as $\{f^+\}$ (Lines 1–2). Then, the algorithm iteratively selects the new feature $f^+$ from the $F \setminus S$ and adds it to the $S$ by selecting the new feature $f^+$ that is determined by Equation (11) (Lines 3–6). The algorithm is terminated when the number of already selected features $|S|$ is equal to the number of features $n$, which is the maximum number of features to be selected, defined by the user. The computational complexity of the Algorithm 1 is $O(n + n^2) = O(n^2)$ because $n$ and $n^2$ unit times are consumed for calculating entropy values of single features and that of feature pairs $f^+$ and $f \in S$.

## 4 EXPERIMENTAL RESULTS

### 4.1 EXPERIMENTAL SETTINGS

To validate the performance of the proposed method, we employed 30 public datasets from two sources: UCI Machine Learning Repository and Kaggle website. The datasets were selected to represent a wide range of domains, including biology, image, game, and audio, and to include various data types, such as numerical and nominal data. Table 2 represents details of employed datasets used in the experiments. The table includes the number of instances $|W|$, original features $|F|$, preprocessed features $|F'|$, and the domain of each dataset. A preprocessing step was applied to each dataset, and each nominal feature in the original feature set $F$ that has more than two categories is converted into the binary features of $F'$ by the one-hot encoding.

Four state-of-the-art UFS methods were selected for performance comparison: EGC [Zhang et al., 2020], U2 [Villa et al., 2021], VCSD [Karami et al., 2023], and FMI [Yuan et al., 2021b]. We detailed the parameter settings for each compared method as follows.

- **EGC** incorporates the between-class scatter matrix and

Table 2: Summary of the datasets used in the experiments

| Dataset | $|W|$ | $|F|$ | $|F'|$ | Domain |
|---|---|---|---|---|
| ALLAML | 72 | 7,129 | 7,129 | Biology |
| Alzheimer | 174 | 450 | 450 | Biology |
| Arcene | 100 | 9,920 | 9,920 | Biology |
| Audiology | 226 | 71 | 93 | Biology |
| Ba | 1,404 | 320 | 320 | Image |
| Chess | 3,196 | 37 | 38 | Game |
| CLL_SUB_111 | 111 | 11,340 | 11,340 | Biology |
| Coil20 | 1,440 | 1,024 | 1,024 | Image |
| Colon | 62 | 2,001 | 5,994 | Biology |
| Leukemia | 72 | 7,070 | 7,070 | Biology |
| LSVT | 126 | 310 | 310 | Biology |
| Lung | 203 | 3,312 | 3,312 | Biology |
| Lymphoma | 96 | 4,026 | 4,026 | Biology |
| Madelon | 2,600 | 500 | 500 | Artificial |
| Mushrooms | 8,124 | 23 | 98 | Biology |
| Nci9 | 60 | 9,712 | 9,712 | Biology |
| Nursery | 12,960 | 9 | 26 | Biology |
| Pdspeech | 756 | 752 | 752 | Audio |
| Promoters | 105 | 58 | 228 | Biology |
| Prostate_GE | 102 | 5,966 | 5,966 | Biology |
| SCADI | 70 | 206 | 206 | Biology |
| Semeion | 1,593 | 256 | 256 | Image |
| SPECT | 265 | 22 | 23 | Biology |
| Splice | 3,190 | 61 | 287 | Biology |
| Tox171 | 171 | 5,748 | 5,748 | Biology |
| Tic-Tac-Toe | 958 | 10 | 27 | Game |
| Umist | 575 | 644 | 644 | Image |
| WarpAR10P | 130 | 2,400 | 2,400 | Image |
| WarpPIE10P | 210 | 2,420 | 2,420 | Image |
| Yaleb | 2,414 | 1,024 | 1,024 | Image |

an adaptive graph structure into the traditional UFS framework. It requires two hyperparameters, $\alpha$ and $\lambda$, which were set to 0.001 and 0.1, respectively.

- **U2** uses a radial basis kernel function to address non-linearity within the conventional UFS framework and does not require any hyperparameters.

- **VCSD** introduces a variance-covariance subspace distance to leverage feature correlations, requiring a hyperparameter $\rho$ to adjust a term in the objective function, which was set to 100.

- **FMI** integrates fuzzy mutual information into the UFS framework and requires a hyperparameter $\lambda$ for fuzzy-based entropy calculations, set to 0.1.

Because the proposed method is based on the entropy calculation that requires the discrete probability distribution of the features, all numerical features are discretized. Specifically, the discretization process is conducted by the equal-width binning method [Talukdar et al., 2018] where the number of bins is set to ten. The maximum number of selected features was set to 300 regarding to conventional UFS setting [Lim

Table 3: Comparison results of five UFS methods in terms of $Entropy$, $PDP$, and the minimum number of features that can ensure all patterns are discriminable.

| Dataset | Entropy | | | | | PDP | | | | | Minimum Number of features | | | | |
|---|---|---|---|---|---|---|---|---|---|---|---|---|---|---|---|
| | Proposed | EGC | U2 | VCSD | FMI | Proposed | EGC | U2 | VCSD | FMI | Proposed | EGC | U2 | VCSD | FMI |
| ALLAML | **6.17** | 6.06 | 6.09 | 6.11 | **6.17** | **1.00** | 0.94 | 0.96 | 0.97 | **1.00** | **4** | 5 | 6 | 5 | **4** |
| Alzheimer | 7.43 | 2.59 | **7.44** | 7.40 | 7.40 | 0.99 | 0.22 | **1.00** | 0.98 | 0.98 | 18 | 71 | **10** | 95 | 23 |
| Arcene | **6.64** | 4.97 | 5.60 | 0.48 | 6.56 | **1.00** | 0.54 | 0.66 | 0.07 | 0.96 | **5** | 26 | 20 | 78 | 10 |
| Audiology | **7.11** | 4.85 | 4.05 | 6.49 | 6.39 | **0.72** | 0.26 | 0.17 | 0.56 | 0.54 | – | – | – | – | – |
| Ba | **10.23** | 7.03 | 6.92 | 8.66 | 7.98 | **0.90** | 0.25 | 0.20 | 0.49 | 0.40 | – | – | – | – | – |
| Chess | 11.63 | 11.49 | 11.54 | **11.64** | 11.63 | 1.00 | 0.92 | 0.95 | **1.00** | 1.00 | 38 | 38 | 37 | **36** | 38 |
| CLL_SUB_111 | **6.79** | 6.74 | 6.78 | 6.74 | **6.79** | **1.00** | 0.97 | 0.99 | 0.97 | **1.00** | **5** | 7 | 7 | 8 | **5** |
| Coil20 | **10.49** | 4.89 | 10.38 | 10.49 | 10.49 | **1.00** | 0.30 | 0.97 | 1.00 | 1.00 | 87 | – | 119 | 107 | 100 |
| Colon | **5.95** | 5.72 | 5.08 | 4.57 | 4.03 | **1.00** | 0.89 | 0.65 | 0.65 | 0.42 | **10** | 17 | 23 | 176 | – |
| Leukemia | **6.17** | 5.98 | 6.09 | 6.11 | 6.11 | **1.00** | 0.90 | 0.96 | 0.97 | 0.97 | **7** | 11 | 20 | 8 | 8 |
| LSVT | **6.98** | 2.10 | 3.24 | 6.13 | 6.91 | **1.00** | 0.09 | 0.31 | 0.67 | 0.97 | **5** | 27 | 42 | 12 | 6 |
| Lung | **7.67** | 5.52 | **7.67** | 7.60 | **7.67** | **1.00** | 0.48 | **1.00** | 0.97 | **1.00** | **5** | 14 | **5** | 7 | **5** |
| Lymphoma | **6.58** | 6.43 | 6.45 | 6.52 | 6.42 | **1.00** | 0.93 | 0.94 | 0.97 | 0.93 | **8** | 11 | 16 | 11 | 14 |
| Madelon | 11.34 | 11.19 | 11.34 | 11.34 | **11.34** | 1.00 | 0.93 | 1.00 | 1.00 | **1.00** | 10 | 14 | 10 | 12 | **9** |
| Mushrooms | **8.49** | 5.42 | 5.42 | 7.26 | 5.22 | **0.09** | 0.01 | 0.01 | 0.03 | 0.01 | – | – | – | – | – |
| Nci9 | **5.91** | 5.74 | 5.81 | 5.59 | 5.73 | **1.00** | 0.92 | 0.95 | 0.85 | 0.92 | **7** | 10 | 9 | 9 | 12 |
| Nursery | 12.71 | **13.66** | 12.08 | **13.66** | 12.26 | 0.60 | **1.00** | 0.33 | **1.00** | 0.40 | 25 | **23** | 25 | **23** | 25 |
| Pdspeech | **9.56** | 0.89 | 9.55 | **9.56** | **9.56** | **1.00** | 0.05 | 0.99 | **1.00** | **1.00** | – | – | – | – | – |
| Promoters | **6.71** | 6.68 | 6.55 | 6.68 | 6.65 | **1.00** | 0.98 | 0.92 | 0.98 | 0.97 | **17** | 22 | 29 | 23 | 33 |
| Prostate_GE | **6.67** | 6.37 | 4.80 | 5.91 | 6.44 | **1.00** | 0.85 | 0.54 | 0.74 | 0.89 | **4** | 8 | 12 | 44 | 9 |
| SCADI | **6.13** | 5.89 | 5.09 | 5.22 | 5.19 | **1.00** | 0.89 | 0.59 | 0.71 | 0.64 | **51** | 98 | 124 | 85 | 116 |
| Semeion | **10.64** | 10.63 | 10.51 | 10.63 | 10.60 | **1.00** | 1.00 | 0.95 | 1.00 | 0.98 | **51** | 71 | 123 | 82 | 97 |
| SPECT | **7.33** | 6.96 | 6.96 | 7.15 | 7.18 | **0.79** | 0.72 | 0.72 | 0.74 | 0.76 | – | – | – | – | – |
| Splice | **11.13** | 9.97 | 7.75 | 11.09 | 11.08 | **0.79** | 0.46 | 0.08 | 0.78 | 0.78 | – | – | – | – | – |
| Tox171 | 9.90 | 8.46 | 8.46 | **9.90** | **9.90** | **1.00** | 0.45 | 0.45 | **1.00** | **1.00** | 17 | 23 | 23 | **17** | **17** |
| Tic-Tac-Toe | **7.42** | 7.34 | 7.29 | 7.35 | 7.37 | **1.00** | 0.96 | 0.95 | 0.96 | 0.98 | **4** | 7 | 5 | 5 | 5 |
| Umist | 9.12 | 8.86 | 9.10 | 9.11 | **9.13** | 0.98 | 0.88 | 0.97 | 0.97 | **0.98** | – | – | – | – | – |
| WarpAR10P | **7.02** | 6.78 | 6.76 | 6.45 | 6.84 | **1.00** | 0.91 | 0.88 | 0.78 | 0.94 | **5** | 10 | 49 | 26 | 122 |
| WarpPIE10P | **7.71** | 7.29 | 6.89 | 7.52 | **7.71** | **1.00** | 0.85 | 0.77 | 0.94 | **1.00** | **10** | 27 | 66 | 84 | **10** |
| Yaleb | **11.12** | 8.54 | 8.72 | 10.55 | 11.06 | **0.97** | 0.70 | 0.71 | 0.88 | 0.96 | – | – | – | – | – |
| Avg. Rank | **1.20** | 3.87 | 3.90 | 2.80 | 2.50 | **1.20** | 3.87 | 3.90 | 2.77 | 2.43 | **1.20** | 2.67 | 2.80 | 2.53 | 2.27 |

and Kim, 2021].

To validate the superiority of UFS methods, three evaluation measures are considered. We measured the entropy of the feature subsets selected by the proposed method and compared methods ($Entropy$), which can be represented as

$$Entropy = H(S). \qquad (13)$$

Next, We employed the pattern discrimination power test ($PDP$) that measures the portion of the discriminable data patterns in the dataset based on the feature subset. The $PDP$ can be represented as

$$PDP(W) = \frac{1}{|W|} \cdot \sum_{i=1}^{|W|} \left[\!\!\left[ \left( \sum_{j=1}^{i-1} [\![ w_i = w_j ]\!] \right) = 0 \right]\!\!\right], \quad (14)$$

where $W$ is the dataset comprised of the feature subset $S$, $w_i$ is the $i$-th pattern, and $[\![\cdot]\!]$ yields one if the proposition stated in the brackets is true and returns zero otherwise. The range of the $PDP$ is from $\frac{1}{|W|}$ to 1, where each value means that all data patterns are indiscriminable/discriminable, respectively. Both $PDP$ and $Entropy$ measures exhibit monotonic increases with the inclusion of additional features [Artstein et al., 2004]. In particular, the relationship between

the $PDP$ and $Entropy$ is illustrated in the supplementary material to show the correlation between the two measures. Finally, the minimum number of features that all patterns become discriminable is measured based on the feature subsets selected by the proposed and compared methods.

## 4.2 EXPERIMENTAL RESULTS

Table 3 presents experimental results on 30 datasets in terms of $Entropy$, $PDP$, and the minimum number of features required to make all patterns discriminable. Because the maximum number of selected features was set to 300, if all patterns within a dataset remain indiscriminable with more than 300 selected features, the corresponding entries in the table are filled with "−" due to the exhaustive time consumption. Subsequently, each entry for $Entropy$ and $PDP$ represents the result corresponding to the smallest number of features required to make all patterns discriminable among all UFS methods. For the datasets where all UFS methods failed to discriminate all patterns with fewer than 300 features, the results for $Entropy$ and $PDP$ are reported when the number of selected features is 17, reflecting the average of the smallest number of features required among the other datasets. In the table, the best results are

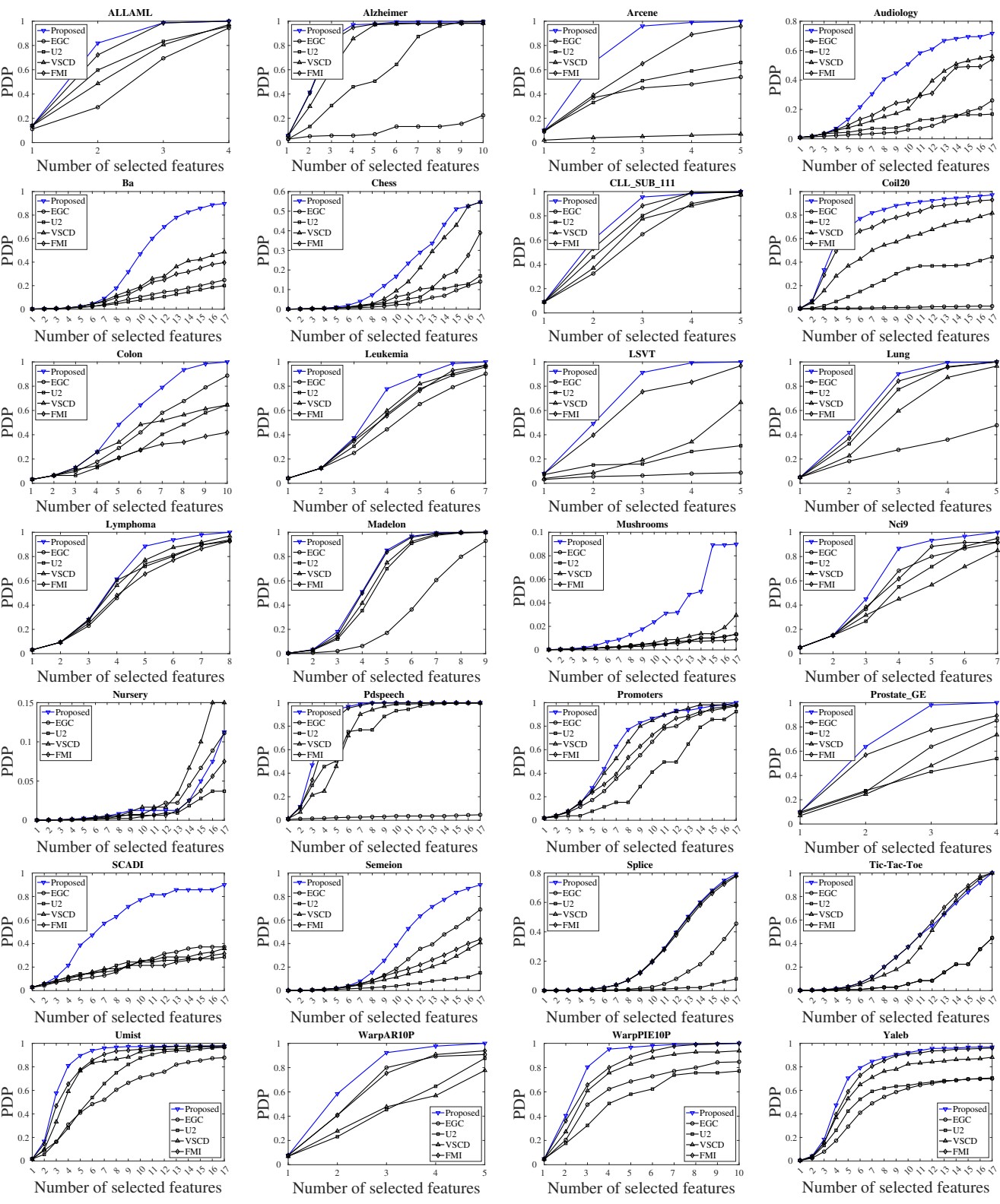

Figure 1: Comparison results of $PDP$ performance according to the number of features selected by the five UFS methods.

Table 4: Comparison results of $Entropy$ and $PDP$ performance based on maximization/minimization of $H(S)$.

| Dataset | Entropy | | PDP | |
| --- | --- | --- | --- | --- |
| | max | min | max | min |
| ALLAML | **6.17** | 0.32 | **1.00** | 0.06 |
| Alzheimer | **7.44** | 2.18 | **1.00** | 0.18 |
| Arcene | **6.62** | 0.08 | **0.99** | 0.02 |
| CLL_SUB_111 | **6.79** | 0.37 | **1.00** | 0.05 |
| Coil20 | **10.49** | 3.67 | **1.00** | 0.25 |
| Colon | **5.95** | 1.81 | **1.00** | 0.19 |
| Leukemia | **6.17** | 0.73 | **1.00** | 0.11 |
| LSVT | **6.96** | 0.13 | **0.99** | 0.02 |
| Lung | **7.67** | 1.20 | **1.00** | 0.14 |
| Nci9 | **5.91** | 0.37 | **1.00** | 0.07 |
| Nursery | **13.66** | 12.66 | **1.00** | 0.50 |
| Prostate_GE | **6.67** | 0.08 | **1.00** | 0.02 |
| Tic-Tac-Toe | **7.42** | 4.52 | **1.00** | 0.26 |
| WarpAR10P | **7.02** | 2.96 | **1.00** | 0.31 |
| WarpPIE10P | **7.70** | 3.67 | **0.99** | 0.28 |
| Avg. Rank | **1.00** | 2.00 | **1.00** | 2.00 |

Table 5: Comparison results of $Entropy$ and $PDP$ performance based on different initial settings, $H(f^+)$ and $H(f^+, f)$, for the proposed method.

| Dataset | Entropy | | PDP | |
| --- | --- | --- | --- | --- |
| | $H(f^+)$ | $H(f^+, f)$ | $H(f^+)$ | $H(f^+, f)$ |
| ALLAML | **6.17** | **6.17** | **1.00** | **1.00** |
| Alzheimer | **7.44** | **7.44** | **1.00** | **1.00** |
| Arcene | 6.62 | **6.64** | 0.99 | **1.00** |
| Audiology | **7.11** | **7.11** | **0.72** | **0.72** |
| Ba | 10.23 | **10.23** | **0.90** | 0.90 |
| Chess | **11.64** | **11.64** | **1.00** | **1.00** |
| CLL_SUB_111 | **6.79** | **6.79** | **1.00** | **1.00** |
| Coil20 | 10.49 | **10.49** | 1.00 | **1.00** |
| Colon | **5.95** | 5.92 | **1.00** | 0.98 |
| Leukemia | **6.17** | 6.14 | **1.00** | 0.99 |
| LSVT | 6.96 | **6.98** | 0.99 | **1.00** |
| Lung | **7.67** | **7.67** | **1.00** | **1.00** |
| Lymphoma | **6.58** | **6.58** | **1.00** | **1.00** |
| Madelon | **11.34** | **11.34** | **1.00** | **1.00** |
| Mushrooms | **8.49** | **8.49** | **0.09** | **0.09** |
| Nci9 | **5.91** | **5.91** | **1.00** | **1.00** |
| Nursery | **13.66** | **13.66** | **1.00** | **1.00** |
| Pdspeech | **9.56** | **9.56** | **1.00** | **1.00** |
| Promoters | **6.71** | **6.71** | **1.00** | **1.00** |
| Prostate_GE | **6.67** | **6.67** | **1.00** | **1.00** |
| SCADI | **6.13** | **6.13** | **1.00** | **1.00** |
| Semeion | **10.64** | **10.64** | **1.00** | **1.00** |
| SPECT | **7.33** | **7.33** | **0.79** | **0.79** |
| Splice | **11.13** | **11.13** | **0.79** | **0.79** |
| Tox171 | **9.90** | **9.90** | **1.00** | **1.00** |
| Tic-Tac-Toe | **7.42** | **7.42** | **1.00** | **1.00** |
| Umist | 9.12 | **9.12** | 0.98 | **0.98** |
| WarpAR10P | **7.02** | **7.02** | **1.00** | **1.00** |
| WarpPIE10P | 7.70 | **7.71** | 0.99 | **1.00** |
| Yaleb | **11.12** | 11.11 | **0.97** | 0.97 |
| Avg. Rank | 1.20 | **1.10** | 1.17 | **1.13** |

highlighted in bold, and we reported the average rank (Avg. Rank) of the proposed and compared methods in the last row of the table.

Experimental results demonstrate that the proposed method outperforms the compared methods in terms of $Entropy$ on 25 out of 30 datasets. These results indicate that the incremental search for the proposed method, along with its derived objective function, effectively selects a feature subset that can maximize the entropy of the datasets with selected features. Consequently, the selected features by the proposed method contain more information content compared to those selected by other methods. Next, the proposed method outperforms the compared methods in terms of $PDP$ on 25 out of 30 datasets, indicating the superiority of the proposed method against the compared methods. In the experiments based on the minimum number of features that can ensure all patterns are discriminable, the superiority of the proposed method is observed again because the proposed method outputs a significantly compact feature subset compared to other methods. Notably, on average, the proposed method selects $31\%$ fewer features than the second-best method on datasets where the proposed method outperforms the compared methods. Overall, the proposed method outperforms compared methods in terms of three pattern discrimination power-related measures: $Entropy$, $PDP$, and the minimum number of features.

Figure 1 illustrates the comparison results of $PDP$ performance as the number of selected features increases until all patterns are discriminable, with a maximum of 17 as same experimental setting in Table 3. Experimental results of 28 out of 30 datasets are represented as line plots where the $x$-

and $y$-axis represent the number of features and the $PDP$ performance, respectively. Figure 1 shows that the proposed method consistently outperformed the compared methods on most of the datasets. Specifically, on Arcene, Audiology, and Coil20 datasets, the proposed method demonstrated superior performance regardless of the number of selected features. Moreover, on SCADI, Semeion, and Lsvt datasets, the $PDP$ performance of the proposed method rapidly increased compared to other methods, indicating the compactness of the feature subset selected by the proposed method.

The proposed method selects a feature that maximizes the entropy, which can be viewed as an opposite concept to the conventional information-theoretic UFS methods. To investigate this aspect, we conducted additional experiments on the maximization and the minimization of the entropy. In contrast to Algorithm 1, the minimization approach starts with a feature with the smallest entropy and then adds a new feature that preserves the entropy of $S$. Table 4 represents the comparison results on the ten datasets. Experimental

results indicate that the two approaches lead to significantly different results. For example, in the case of the Coil20 dataset, the feature subset based on the maximization approach yields a $1.00$ $PDP$ value, indicating that all patterns are discriminable. In contrast, $75\%$ of patterns are indiscriminable when the minimization approach is applied because the corresponding $PDP$ value is $0.25$.

In Section 3.2, we explained that the algorithm may start with either maximizing $H(f^+)$ or $H(f^+, f)$ where $f \in F \setminus f^+$. To clarify whether there is a significant difference according to the initial setting, we conducted additional experiments because it affects the entire subsequent iterations. Table 5 shows the experimental results of different initial settings in terms of $Entropy$ and $PDP$. Overall, the results for $H(f^+)$ and $H(f^+, f)$ are largely consistent with one another, indicating that the initial setting affects the FS process insignificantly in terms of $Entropy$ and $PDP$.

Finally, we evaluate the classification accuracy and the MI based on the feature subset because it was frequently used in traditional UFS studies. The experimental results show that the feature subset identified by the proposed method can yield better classification results. In addition, we visualize the entropy value according to the number of features selected by the proposed and compared methods. Detailed experimental results are provided in the supplementary material.

## 5 CONCLUSION

In this work, we proposed a UFS method based on maximizing entropy. This approach aims to produce a subset of features that maximizes the discriminability among patterns, thus serving the need for identifying novel patterns. With a simple example demonstrating the consequences of the minimization and maximization approaches, we provided a rigorous formulation of the score function based on theoretical derivation. The experimental results showed that the proposed method outperforms compared methods in terms of pattern discrimination power-related measures.

Future work can explore the potential for real-world applications of the proposed method, including but not limited to, real-time recommendation systems, search engines, and dynamic content optimization. Moreover, the proposed method could be extended to handle different types of data and computational frameworks, providing a more universal solution to the challenges of high-dimensional data.

### Acknowledgements

This work was supported by Institute of Information & communications Technology Planning & Evaluation (IITP) grant funded by the Korea government(MSIT) (2021-0-01341,Artificial Intelligence Graduate School Program(Chung-Ang University)).

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

# Unsupervised Feature Selection towards Maximizing Pattern Discrimination Power (Supplementary Material)

**Wangduk Seo**[1]                     **Jaesung Lee**[1,2‡]

[1]AI/ML Innovation Research Center, Chung-Ang University, Seoul, South Korea
[2]Department of Artificial Intelligence, Chung-Ang University, Seoul, South Korea

## A   RELATIONSHIP BETWEEN $PDP$ AND $Entropy$ MEASURES

In this section, we provide a theoretical proof to establish the relationship between the $Entropy$, the joint entropy of a feature subset, and the $PDP$ measure. $PDP$ is defined as

$$PDP(W) = \frac{1}{|W|} \cdot \sum_{i=1}^{|W|} \left[\!\!\left[ \left( \sum_{j=1}^{i-1} [\![w_i = w_j]\!] \right) = 0 \right]\!\!\right], \tag{15}$$

where $W$ is the dataset comprised of the feature subset $S$, $w_i$ is the $i$-th pattern, and $[\![\cdot]\!]$ yields one if the proposition stated in the brackets is true and returns zero otherwise.

We first provide a proposition to establish the relationship between the joint entropy of a feature subset and the $PDP$ measure.

**Proposition 2.** *The joint entropy $H(S)$ is maximized when all patterns in the dataset $W$ are discriminable.*

*Proof.* The joint entropy $H(S)$ of the dataset quantifies the level of uncertainty within the feature subset $S$. It is defined as $H(S) = -\sum_{i=1}^{|W|} P(w_i) \log P(w_i)$, where $P(w_i)$ represents the probability of the $i$-th pattern occurring in the dataset. To simplify, let $P(w_i) = y_i$, transforming our expression to function terms as $f(y) = -y \log y$ with $y = P(w_i)$.

Considering the concavity of the function $f(y) = -y \log y$, we can employ Jensen's Inequality [Jensen, 1906] to establish an upper bound for the sum $\sum_{i=1}^{|W|} f(y_i)$, which is essential for our entropy calculation as

$$\sum_{i=1}^{|W|} f(P(w_i)) = \sum_{i=1}^{|W|} -P(w_i) \log P(w_i) \le |W| \cdot f\left(\frac{1}{|W|}\right). \tag{16}$$

This formulation bounds the joint entropy as

$$-\sum_{i=1}^{|W|} P(w_i) \log P(w_i) \le \log |W|. \tag{17}$$

When all patterns in $W$ are discriminable, this condition indicates a uniform distribution of occurrences, with $P(w_i) = \frac{1}{|W|}$ for all $i$, thereby transforming our inequality into the equality as

$$H(S) = \log |W|. \tag{18}$$

---

[*]Corresponding author
[‡]Corresponding author

*Accepted for the 40$^{th}$ Conference on Uncertainty in Artificial Intelligence* (UAI 2024).

Thus, when all patterns in the dataset $W$ are discriminable, the joint entropy $H(S)$ of the dataset is maximized. □

Because the proposed score function aims to maximize $H(S)$, a theoretical proof is provided to establish the relationship between the entropy of a feature subset and $PDP$, showing that a decrease in $PDP$ leads to a decrease in the upper bound of $H(S)$. First, let $m = (1 - PDP(W)) \cdot |W|$. For instance, consider $m = 4$ and $|W| = 10$ with a $PDP$ value of 0.6. In this case, at least five patterns are indiscriminable from each other, as illustrated by the sequence $\{1, 2, 3, 4, 5, 6, 6, 6, 6, 6\}$, where the last five patterns are indiscriminable. Conversely, the situation with a maximum of eight indiscriminable patterns within the dataset could be exemplified by $\{1, 2, 3, 3, 4, 4, 5, 5, 6, 6\}$. Based on the example, the upper bound of $H(S)$ can be represented by the following lemma.

**Lemma 2.** *The upper bound joint entropy $H(S)$ of a dataset $W$ is bounded as*

$$H(S) \leq -\frac{|W| - 2m}{|W|} \cdot \log \frac{1}{|W|} - \frac{2m}{|W|} \cdot \log \frac{2}{|W|}, \tag{19}$$

*where $m = (1 - PDP(W)) \cdot |W|$.*

*Proof.* The number of indiscriminable patterns in $W$ ranges from $m + 1$ to $2m$, as illustrated in the provided example. Thus, the entropy of $|W| - 2m$ discriminable patterns, $-\frac{|W|-2m}{|W|} \cdot \log \frac{1}{|W|}$, is constant. Considering the remaining $2m$ patterns, the entropy is maximized when the patterns are uniformly distributed, as mentioned in Proposition 2. This condition is achieved when there are $m$ pairs of patterns, with each pair being identical, yielding $-\frac{2m}{|W|} \cdot \log \frac{2}{|W|}$. Equation (19) represents the upper bound of the joint entropy $H(S)$. □

According to Lemma 2, the relationship between the $PDP$ and the joint entropy $H(S)$ can be established, demonstrating that a decrease in $PDP$, which corresponds to an increase in $m$, leads to a decrease in the upper bound of $H(S)$ Given the uncertainty regarding the precise number of indiscriminable patterns as $m$ increases, we construct our proof by focusing on the upper bounds of $H(S)$ for both $m$ and $m + 1$.

**Theorem 2.** *The upper bound of $H(S)$ for $m$ is greater than the upper bound of $H(S)$ for $m + 1$.*

*Proof.* By applying Lemma 2, the upper bound of $H(S)$ for $m + 1$ represents as follows:

$$-\frac{|W| - 2(m+1)}{|W|} \cdot \log \frac{1}{|W|} - \frac{2(m+1)}{|W|} \cdot \log \frac{2}{|W|} \tag{20}$$

Subtracting the upper bound of $H(S)$ for $m + 1$ from the upper bound of $H(S)$ for $m$, as detailed in Equation 19, we obtain

$$\begin{aligned}
&-\frac{|W| - 2m}{|W|} \cdot \log \frac{1}{|W|} - \frac{2m}{|W|} \cdot \log \frac{2}{|W|} \\
&-\left( -\frac{|W| - 2(m+1)}{|W|} \cdot \log \frac{1}{|W|} - \frac{2(m+1)}{|W|} \cdot \log \frac{2}{|W|} \right) \\
&= \frac{2}{|W|} \geq 0.
\end{aligned} \tag{21}$$

□

Because the $PDP$ and the upper bound of $H(S)$ are both decreasing functions of $m$, Theorem 2 indicates that the proposed method, which aims to maximize $H(S)$, can effectively select a feature subset that maximizes the pattern discrimination power.

# B  ADDITIONAL EXPERIMENTAL RESULTS

In Table 6, we present a comparative analysis of the execution times and MI for the conducted UFS methods. The best performance is highlighted in bold, and the average rank is presented at the bottom of the table. Execution times are reported in seconds, reflecting the computational effort required to process each dataset. In particular, the experiments were conducted on a system equipped with a 13th Generation Intel® Core™ i9-13900K processor, clocked at 3.00 GHz. As a

Table 6: Comparsion results of execution time and MI performance.

| Dataset | Execution Time in Seconds | | | | | MI | | | | |
|---|---|---|---|---|---|---|---|---|---|---|
| | Proposed | EGC | U2 | VCSD | FMI | Proposed | EGC | U2 | VCSD | FMI |
| ALLAML | 375.65 | **221.69** | 5978.10 | 1138.32 | 1486.96 | **0.93** | 0.46 | 0.54 | 0.04 | 0.50 |
| Alzheimer | 1165.01 | **662.63** | 23696.24 | 3816.60 | 5357.24 | **0.97** | 0.03 | 0.08 | 0.34 | 0.71 |
| Arcene | 303.85 | **149.41** | 4204.49 | 741.45 | 1587.23 | **0.95** | 0.34 | 0.22 | 0.06 | 0.17 |
| Audiology | 0.18 | **0.01** | 1.96 | 0.46 | 0.74 | **3.04** | 1.32 | 2.08 | 2.03 | 2.01 |
| Ba | 0.06 | 0.06 | 1.05 | **0.04** | 0.17 | **4.17** | 1.60 | 1.41 | 2.97 | 3.14 |
| Chess | 2.28 | **0.10** | 7.33 | 3.87 | 13.82 | **0.58** | 0.32 | 0.26 | 0.26 | 0.22 |
| CLL_SUB_111 | 775.53 | **440.78** | 15112.69 | 2839.68 | 3555.26 | **1.37** | 0.93 | 0.87 | 0.41 | **1.37** |
| Coil20 | 0.08 | **0.07** | 0.07 | 0.27 | 4.89 | **4.32** | 0.57 | 2.87 | 4.32 | 4.27 |
| Colon | 7.83 | **3.59** | 16.22 | 6.59 | 650.64 | **0.94** | 0.87 | 0.81 | 0.66 | 0.41 |
| Leukemia | **0.20** | 27.34 | 9.36 | 12.26 | 514.60 | **0.93** | **0.93** | **0.93** | 0.88 | 0.76 |
| Lsvt | 92.26 | **4.93** | 49.98 | 20.90 | 8496.83 | **0.89** | 0.05 | 0.05 | 0.41 | 0.13 |
| Lung | 266.99 | **136.40** | 3663.49 | 768.40 | 911.85 | **1.49** | 1.38 | **1.49** | **1.49** | **1.49** |
| Lymphoma | 388.49 | **218.35** | 5872.40 | 1102.46 | 1600.35 | **2.45** | **2.45** | **2.45** | **2.45** | **2.45** |
| Madelon | 0.99 | **0.15** | 6.03 | 2.46 | 4.38 | **0.35** | 0.00 | 0.23 | 0.01 | 0.24 |
| Mushrooms | 162.93 | **22.47** | 569.18 | 166.40 | 1085.49 | **0.93** | 0.76 | 0.76 | 0.67 | 0.86 |
| Nci9 | 148.46 | **40.09** | 954.09 | 263.30 | 660.76 | **3.08** | 2.64 | **3.08** | 0.58 | 2.98 |
| Nursery | 34.69 | 21.69 | 60.47 | **19.09** | 12031.08 | **1.13** | 0.17 | 0.38 | 1.06 | 1.11 |
| Pdspeech | **2.06** | 32.62 | 28.36 | 52.79 | 5279.50 | **0.65** | 0.01 | 0.09 | 0.10 | 0.14 |
| Promoters | 659.48 | **441.33** | 16076.21 | 2555.64 | 3328.74 | **1.00** | 0.71 | 0.41 | 0.97 | 0.96 |
| Prostate_GE | **0.40** | 1389.47 | 184.80 | 194.09 | 22083.95 | **0.98** | 0.71 | 0.86 | 0.15 | 0.20 |
| SCADI | 18.67 | **1.17** | 19.37 | 10.07 | 712.49 | **2.18** | 1.66 | 1.62 | 1.63 | 1.54 |
| Semeion | 0.51 | **0.03** | 3.97 | 1.28 | 1.85 | **2.62** | 1.94 | 0.82 | 1.29 | 2.13 |
| SPECT | 5.58 | **3.64** | 11.89 | 5.70 | 992.32 | 0.40 | 0.37 | 0.37 | **0.42** | 0.39 |
| Splice | **14.00** | 29.64 | 55.27 | 21.94 | 7801.44 | **1.32** | 0.03 | 0.02 | 1.27 | 1.31 |
| Tox171 | 426.87 | **120.82** | 3212.45 | 637.41 | 2417.68 | **2.00** | 1.97 | **2.00** | 1.90 | **2.00** |
| TTTgame | **0.04** | 0.66 | 1.16 | 0.68 | 5.77 | **0.93** | 0.19 | 0.19 | 0.62 | **0.93** |
| Umist | 16.96 | **0.59** | 14.52 | 7.02 | 479.49 | **4.28** | 4.27 | **4.28** | **4.28** | **4.28** |
| WarpAR10P | 68.45 | **7.59** | 191.55 | 78.93 | 264.45 | **3.32** | 3.25 | 3.17 | 3.18 | 3.20 |
| WarpPIE10P | 99.19 | **7.64** | 207.88 | 80.65 | 491.89 | **3.32** | **3.32** | 3.02 | 3.14 | **3.32** |
| Yaleb | 185.28 | **18.93** | 127.98 | 31.47 | 35409.53 | **5.24** | 5.18 | 4.30 | 5.13 | 5.23 |
| Avg. Rank | 2.20 | **1.47** | 4.07 | 2.70 | 4.57 | **1.03** | 3.47 | 3.33 | 3.33 | 2.60 |

result, our proposed method ranks as the second-fastest method on average, following EGC, which highlights its efficiency and potential applicability to a wide range of datasets. Furthermore, for algorithms such as EGC, which rely on specific parameter settings, it is important to note that the time to achieve optimal results can vary significantly depending on the number of parameters adjusted. In such contexts, the execution time can increase substantially based on the complexity of the parameter space being navigated.

Subsequently, in traditional UFS studies, the superiority of devised methods was validated by measuring the MI between selected feature subset and class variable or directly using classifiers such as naïve Bayes or decision tree. Although this kind of validation is only possible when the dataset contains the class variable $C$, it is reasonable from the viewpoint of information theory because the increment of MI between input features and class variable leads to the reduction of naïve Bayes error owing to the Fano's inequality and $k$-cardinality approximation of MI between $S$ and $C$ when $k = 1$ leads to the unsupervised score function [Seo et al., 2019], i.e., maximizing $H(S)$ can contribute to the increment of MI $M(S, C) = H(S) + H(C) - H(S, C)$. Here, we report the performance of the proposed and compared methods in terms of MI and classification accuracy.

In Table 6, the proposed method achieved the best performance on 29 of the 30 datasets, with an average rank of 1.03. These results suggest that the proposed method could potentially improve classification performance more effectively than the compared methods. To verify this, experiments regarding the classification accuracy were performed using the features selected by the proposed and compared methods.

Table 7: Comparison results of classification accuracy performance based on the feature subsets selected by the five UFS methods.

| Dataset | Naïve Bayes | | | | | Decision Tree | | | | |
|---|---|---|---|---|---|---|---|---|---|---|
| | Proposed | EGC | U2 | VCSD | FMI | Proposed | EGC | U2 | VCSD | FMI |
| ALLAML | 0.57 ± 0.22 | **0.72 ± 0.20** | 0.67 ± 0.11 | 0.65 ± 0.24 | 0.71 ± 0.10 | 0.56 ± 0.23 | **0.68 ± 0.21** | 0.61 ± 0.13 | 0.65 ± 0.24 | 0.59 ± 0.19 |
| Alzheimer | **0.76 ± 0.12** | 0.41 ± 0.08 | 0.52 ± 0.14 | 0.71 ± 0.09 | 0.68 ± 0.13 | **0.75 ± 0.09** | 0.42 ± 0.09 | 0.45 ± 0.13 | 0.65 ± 0.08 | 0.69 ± 0.12 |
| Arcene | 0.60 ± 0.14 | 0.45 ± 0.16 | **0.62 ± 0.15** | 0.56 ± 0.14 | 0.58 ± 0.12 | 0.58 ± 0.08 | 0.49 ± 0.09 | 0.56 ± 0.15 | 0.55 ± 0.14 | **0.60 ± 0.12** |
| Audiology | **0.63 ± 0.14** | 0.52 ± 0.10 | 0.50 ± 0.10 | 0.35 ± 0.12 | 0.30 ± 0.12 | **0.66 ± 0.10** | 0.52 ± 0.11 | 0.49 ± 0.10 | 0.33 ± 0.11 | 0.38 ± 0.12 |
| Ba | **0.25 ± 0.03** | 0.07 ± 0.02 | 0.07 ± 0.03 | 0.21 ± 0.03 | 0.24 ± 0.05 | 0.21 ± 0.04 | 0.08 ± 0.02 | 0.08 ± 0.02 | 0.21 ± 0.03 | **0.25 ± 0.03** |
| Chess | **0.75 ± 0.03** | 0.71 ± 0.02 | 0.67 ± 0.01 | 0.63 ± 0.02 | 0.62 ± 0.03 | **0.82 ± 0.02** | 0.73 ± 0.02 | 0.70 ± 0.02 | 0.67 ± 0.01 | 0.66 ± 0.02 |
| CLL_SUB_111 | 0.62 ± 0.19 | 0.32 ± 0.12 | 0.46 ± 0.12 | 0.57 ± 0.16 | **0.65 ± 0.16** | 0.56 ± 0.18 | 0.38 ± 0.14 | 0.33 ± 0.15 | **0.59 ± 0.12** | 0.49 ± 0.13 |
| Coil20 | **0.85 ± 0.03** | 0.10 ± 0.02 | 0.36 ± 0.05 | 0.78 ± 0.04 | 0.82 ± 0.04 | 0.81 ± 0.04 | 0.10 ± 0.02 | 0.45 ± 0.03 | 0.79 ± 0.03 | **0.81 ± 0.04** |
| Colon | 0.57 ± 0.20 | 0.53 ± 0.22 | **0.66 ± 0.22** | 0.59 ± 0.15 | 0.53 ± 0.22 | 0.42 ± 0.15 | 0.50 ± 0.24 | 0.44 ± 0.21 | 0.32 ± 0.10 | **0.60 ± 0.13** |
| Leukemia | **0.80 ± 0.15** | 0.54 ± 0.17 | 0.67 ± 0.19 | 0.65 ± 0.14 | 0.69 ± 0.15 | 0.44 ± 0.04 | 0.13 ± 0.02 | 0.28 ± 0.03 | 0.29 ± 0.04 | **0.47 ± 0.04** |
| LSVT | **0.77 ± 0.06** | 0.66 ± 0.15 | 0.66 ± 0.15 | 0.70 ± 0.12 | 0.60 ± 0.10 | **0.73 ± 0.13** | 0.48 ± 0.22 | 0.63 ± 0.19 | 0.69 ± 0.12 | 0.67 ± 0.16 |
| Lung | 0.80 ± 0.07 | 0.74 ± 0.08 | 0.75 ± 0.08 | 0.80 ± 0.08 | **0.86 ± 0.06** | 0.69 ± 0.11 | 0.66 ± 0.15 | 0.67 ± 0.15 | **0.79 ± 0.07** | 0.60 ± 0.13 |
| Lymphoma | 0.58 ± 0.18 | 0.51 ± 0.22 | 0.43 ± 0.11 | 0.75 ± 0.15 | **0.80 ± 0.13** | 0.80 ± 0.12 | 0.66 ± 0.10 | 0.62 ± 0.09 | 0.76 ± 0.07 | **0.83 ± 0.08** |
| Madelon | 0.56 ± 0.02 | 0.47 ± 0.02 | 0.49 ± 0.02 | 0.47 ± 0.02 | **0.62 ± 0.03** | 0.50 ± 0.17 | 0.47 ± 0.22 | 0.48 ± 0.16 | 0.58 ± 0.16 | **0.64 ± 0.13** |
| Mushrooms | 0.89 ± 0.02 | **0.93 ± 0.01** | **0.93 ± 0.01** | 0.80 ± 0.02 | 0.90 ± 0.03 | 0.54 ± 0.02 | 0.47 ± 0.02 | 0.51 ± 0.03 | 0.47 ± 0.02 | **0.69 ± 0.03** |
| Nci9 | **0.17 ± 0.16** | 0.07 ± 0.09 | **0.17 ± 0.14** | 0.10 ± 0.09 | **0.17 ± 0.16** | **0.98 ± 0.00** | 0.94 ± 0.01 | 0.94 ± 0.01 | 0.86 ± 0.01 | 0.96 ± 0.01 |
| Nursery | 0.76 ± 0.01 | 0.42 ± 0.01 | 0.51 ± 0.01 | **0.76 ± 0.01** | 0.76 ± 0.01 | 0.18 ± 0.12 | 0.10 ± 0.12 | **0.20 ± 0.15** | 0.15 ± 0.12 | 0.13 ± 0.17 |
| Pdspeech | **0.75 ± 0.03** | **0.75 ± 0.03** | 0.74 ± 0.03 | 0.74 ± 0.04 | 0.69 ± 0.06 | **0.79 ± 0.01** | 0.41 ± 0.01 | 0.54 ± 0.01 | 0.76 ± 0.01 | 0.78 ± 0.01 |
| Promoters | **0.93 ± 0.11** | 0.67 ± 0.19 | 0.48 ± 0.16 | 0.92 ± 0.13 | 0.88 ± 0.13 | 0.37 ± 0.07 | 0.36 ± 0.05 | 0.15 ± 0.07 | 0.35 ± 0.08 | **0.45 ± 0.06** |
| Prostate_GE | 0.74 ± 0.10 | **0.79 ± 0.14** | 0.64 ± 0.13 | 0.57 ± 0.15 | 0.57 ± 0.24 | 0.70 ± 0.08 | 0.75 ± 0.03 | 0.74 ± 0.03 | 0.75 ± 0.03 | **0.79 ± 0.04** |
| SCADI | 0.80 ± 0.17 | 0.66 ± 0.14 | **0.80 ± 0.15** | 0.74 ± 0.11 | 0.73 ± 0.16 | **0.90 ± 0.13** | 0.72 ± 0.13 | 0.53 ± 0.12 | 0.88 ± 0.11 | 0.86 ± 0.13 |
| Semeion | 0.50 ± 0.03 | 0.53 ± 0.07 | 0.28 ± 0.03 | 0.27 ± 0.04 | **0.54 ± 0.04** | 0.72 ± 0.13 | **0.79 ± 0.13** | 0.54 ± 0.17 | 0.61 ± 0.13 | 0.55 ± 0.20 |
| SPECT | **0.81 ± 0.06** | 0.76 ± 0.06 | 0.76 ± 0.06 | 0.81 ± 0.05 | 0.80 ± 0.07 | 0.80 ± 0.15 | 0.73 ± 0.18 | 0.80 ± 0.15 | 0.76 ± 0.17 | 0.77 ± 0.10 |
| Splice | **0.93 ± 0.02** | 0.52 ± 0.02 | 0.52 ± 0.02 | 0.91 ± 0.02 | 0.91 ± 0.02 | 0.53 ± 0.05 | 0.54 ± 0.07 | 0.31 ± 0.04 | 0.28 ± 0.03 | **0.56 ± 0.05** |
| Tox171 | 0.45 ± 0.15 | 0.30 ± 0.08 | 0.46 ± 0.11 | **0.51 ± 0.15** | 0.47 ± 0.08 | 0.81 ± 0.08 | 0.80 ± 0.08 | 0.80 ± 0.08 | **0.84 ± 0.08** | 0.81 ± 0.07 |
| Tic-Tac-Toe | 0.72 ± 0.06 | 0.65 ± 0.04 | 0.65 ± 0.04 | 0.69 ± 0.05 | **0.72 ± 0.05** | 0.93 ± 0.02 | 0.52 ± 0.02 | 0.52 ± 0.02 | 0.93 ± 0.01 | **0.93 ± 0.01** |
| Umist | 0.63 ± 0.08 | 0.69 ± 0.06 | 0.83 ± 0.03 | 0.60 ± 0.07 | **0.83 ± 0.07** | 0.52 ± 0.12 | 0.36 ± 0.11 | 0.44 ± 0.13 | 0.40 ± 0.06 | 0.48 ± 0.11 |
| WarpAR10P | 0.40 ± 0.14 | 0.18 ± 0.09 | 0.27 ± 0.15 | 0.26 ± 0.17 | **0.46 ± 0.15** | 0.94 ± 0.02 | 0.69 ± 0.04 | 0.69 ± 0.04 | 0.82 ± 0.04 | **0.95 ± 0.02** |
| WarpPIE10P | **0.49 ± 0.11** | 0.31 ± 0.06 | 0.45 ± 0.10 | 0.24 ± 0.09 | 0.37 ± 0.10 | 0.64 ± 0.10 | 0.69 ± 0.03 | **0.80 ± 0.07** | 0.67 ± 0.05 | 0.77 ± 0.08 |
| Yaleb | 0.08 ± 0.02 | **0.09 ± 0.02** | 0.06 ± 0.01 | 0.07 ± 0.02 | 0.07 ± 0.01 | 0.38 ± 0.08 | 0.20 ± 0.07 | 0.33 ± 0.13 | 0.41 ± 0.19 | **0.47 ± 0.09** |
| Avg. Rank | **1.97** | 3.70 | 3.17 | 3.30 | 2.57 | **2.09** | 3.73 | 3.61 | 3.09 | 2.27 |

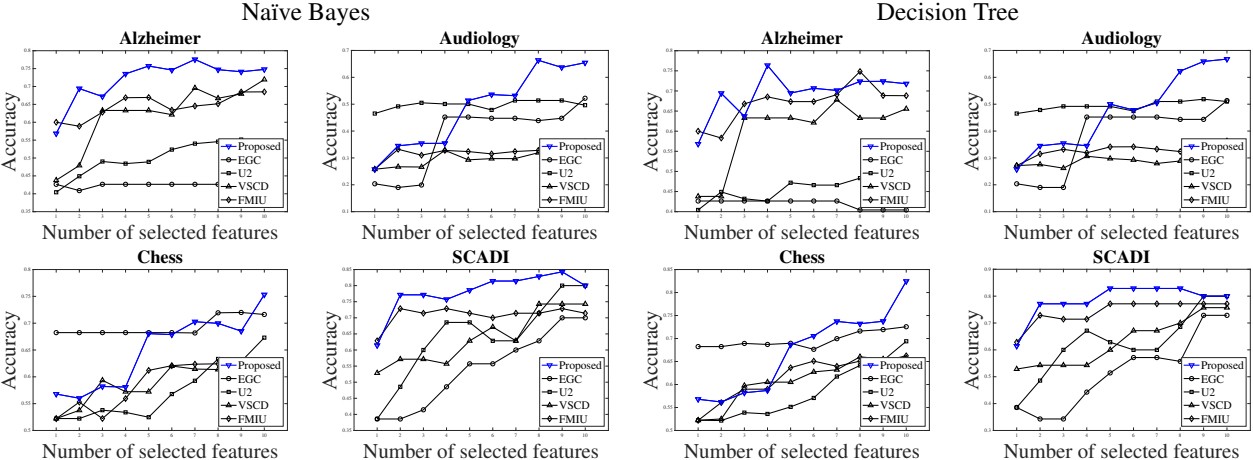

Figure 2: Comparison results of classification accuracy performance according to the number of features selected by the five UFS methods

Table 7 summarizes the classification accuracy of features selected by both the proposed and compared methods, evaluated using naïve Bayes and decision tree classifiers. Specifically, the classification accuracy is measured by the 10-fold cross-validation with the naïve Bayes classifier and the decision tree classifier, which were trained by the data composed of the selected features. In the case of the naïve Bayes classifier, the proposed method achieved superior classification accuracy on 14 out of 30 datasets, with an average rank of 1.97. This performance outperformed that of the next most effective method, FMI, which garnered an average rank of 2.57. For the decision tree classifier, the proposed method also led the field, securing the highest classification accuracy on 9 out of the 30 datasets and an average rank of 2.09. This surpassed the second-best method, EGC, which obtained an average rank of 2.27.

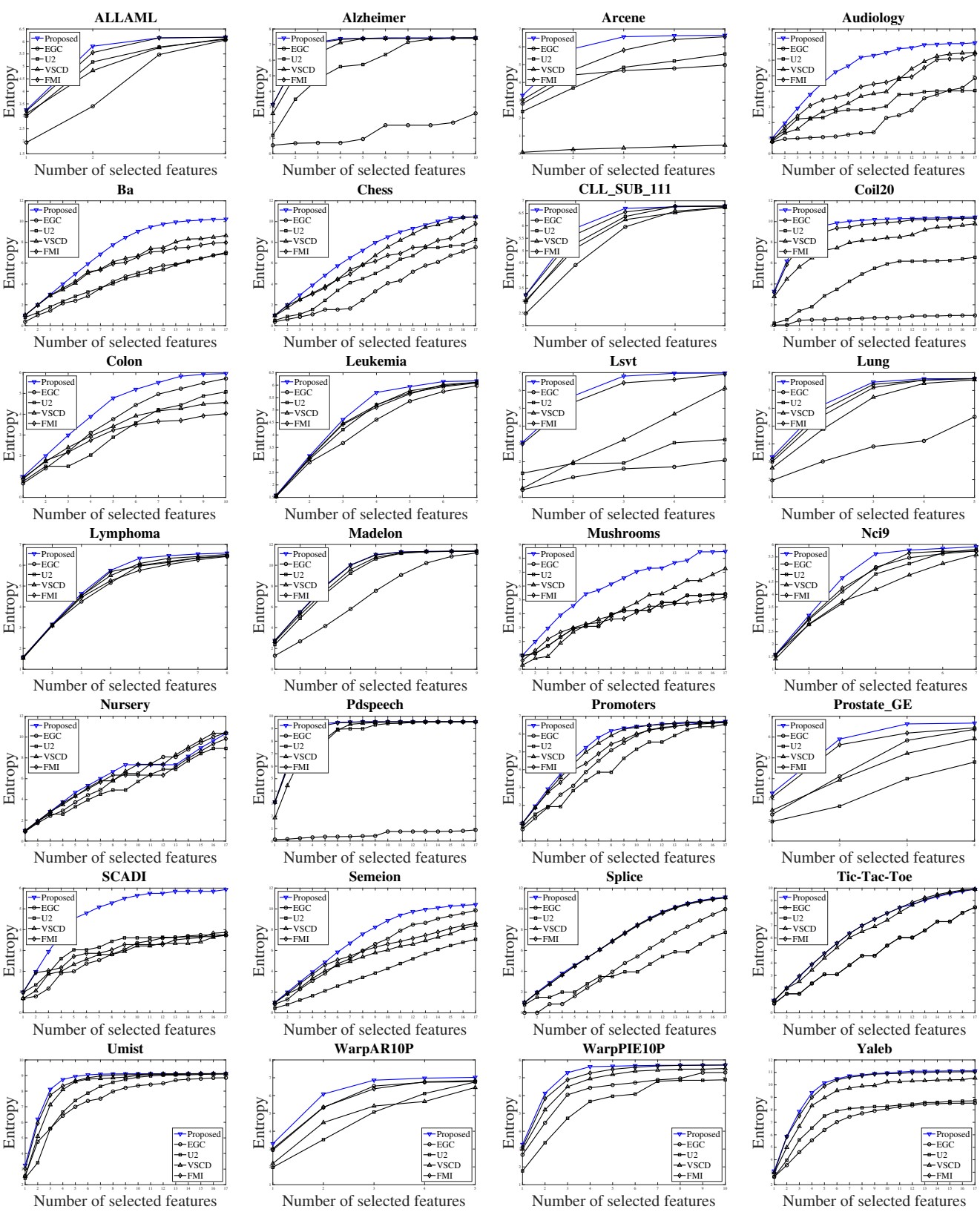

Figure 3: Comparison results of entropy performance according to the number of features selected by the five UFS methods

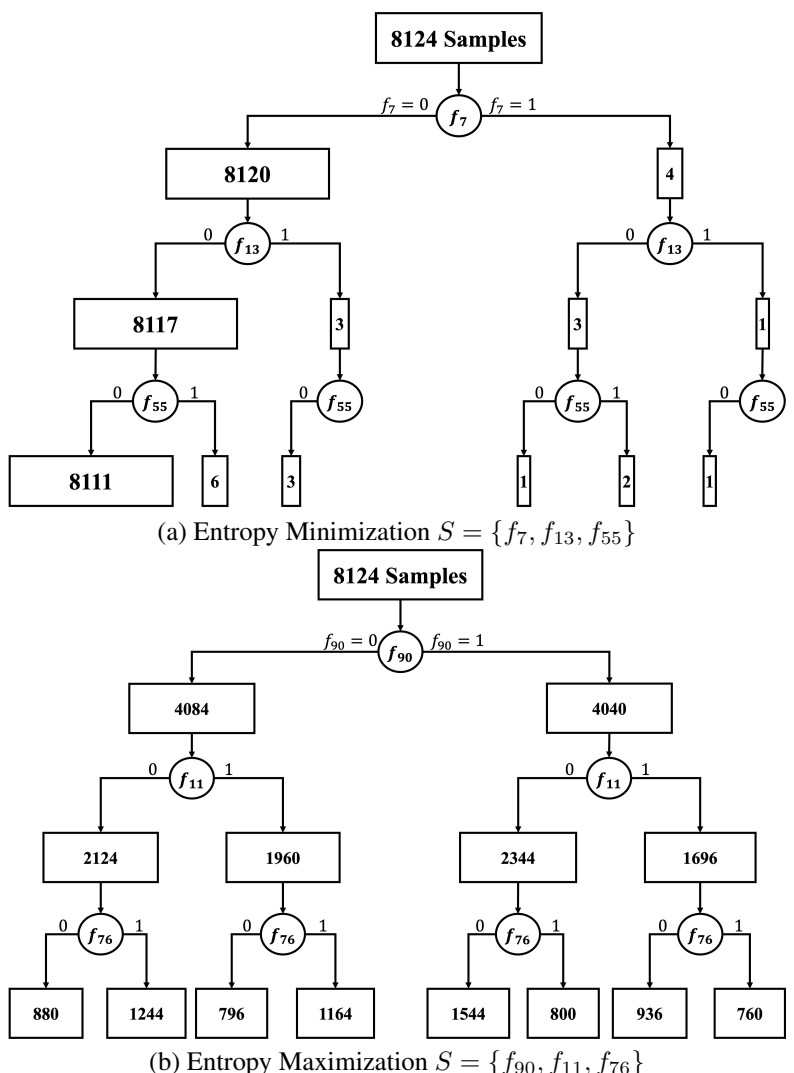

(a) Entropy Minimization $S = \{f_7, f_{13}, f_{55}\}$

(b) Entropy Maximization $S = \{f_{90}, f_{11}, f_{76}\}$

Figure 4: Taxonomic trees constructed by selected features from the proposed method (Entropy Maximization) and the entropy minimization method.

Figure 2 illustrates the classification accuracy achieved by the proposed and compared methods across varying feature sizes on the Alzheimer, Audiology, Chess, and SCADI datasets. In the figure, the blue line symbolizes the proposed method, while the black line represents the comparative methods. Notably, the classification accuracy of the proposed method surpassed that of the compared methods across all four datasets as the feature size increased. This superior performance can be attributed to the feature subset that is optimized towards pattern discrimination power, thereby contributing to improved classification accuracy.

Figure 3 shows the comparison result of entropy performance according to the number of features selected by the five UFS methods. Although it is a trivial result because the proposed method directly maximizes the entropy of the feature subset, whereas compared methods do not, we observed that the proposed method outperforms the compared method significantly in terms of entropy.

## C APPLICATION: TAXONOMIC TREE CONSTRUCTION

The motivation behind maximizing entropy is to fully utilize the discrimination power of patterns, leading to improved information retrieval [Zhong et al., 2011], such as enhancing the process of taxonomy construction for each pattern. This method aligns with information theory principles by selecting features that offer maximal informational value, thereby

---

**Algorithm 2** Incremental Search for the Proposed Method when $k = 3$

---

1: $f^+ \leftarrow \arg\max_{f^+ \in F} H(f^+)$                      ▷ Select the first feature

2: $S \leftarrow \{f^+\}$

3: $f^+ \leftarrow \arg\max_{f^+ \in F-S} \sum_{f \in S} H(f^+, f)$           ▷ Select the second feature

4: $S \leftarrow \{f^+\}$

5: **while** $|S| < n$ **do**

6:      $f^+ \leftarrow \arg\max_{f^+ \in F-S} \sum_{f_i \in S} \sum_{f_j \in S} H(f^+, f_i, f_j)$

7:      $S \leftarrow S \cup \{f^+\}$

8: **end while**

---

improving pattern discrimination power. Specifically, constructing the taxonomic tree [Reiman et al., 2020] starting from the root node and distributing patterns into the tree based on prioritized features with the highest entropy enables a significant reduction in the maximum depth of the tree. This effectiveness arises because features selected in order of descending entropy are arranged from the root node downwards, allowing the values of each pattern associated with these features to be evenly distributed across the tree. Such an organization ensures that the tree expands in a balanced manner, simplifies analytical processes by making the data structure more compact, and enhances the clarity and efficiency of data interpretation. The proposed method selects features iteratively based on their scores from the score function that maximizes the entropy of the feature subset. Considering that the highest entropy is observed in a uniform distribution when patterns are distributed using the selected features in a tree structure, this strategy enables the construction of a tree that closely approximates a balanced tree.

We conducted experiments on the binary Mushrooms dataset to assess the effectiveness of taxonomic trees constructed via the proposed entropy maximization method compared to the conventional entropy minimization approach. For the entropy minimization method, features were selected using the same algorithm as the proposed method, except the objective was to minimize entropy within the feature subset. Both strategies selected three features from the dataset and constructed the trees starting from the root node in the selection order, resulting in trees with a depth of three levels. Patterns were then assigned to each node based on their feature values.

Figure 4 depicts the constructed taxonomic trees. The numbers in nodes indicate the number of assigned patterns at each node, whereas the circles highlight the feature at each division. To visually represent the quantity of patterns per node, the width of each node was adjusted logarithmically in proportion to the number of patterns it contains. The tree derived from the entropy minimization method revealed a significantly skewed structure, with the majority of patterns assigned in the leftmost node. In contrast, the tree resulting from the proposed entropy maximization method exhibited a more balanced structure, approximating a uniform distribution of patterns across the nodes. Given the skewed nature of the tree resulting from the entropy minimization method, which may frequently require additional comparisons to locate a novel pattern, the proposed entropy maximization method presents itself as a potentially more efficient solution, such as information retrieval systems.

## D  SCORE FUNCTION VARYING $k$-CARDINALITY

We provide a detailed explanation of the implications of using values of $k$ larger than 2 by giving a concrete example by introducing a newly instantiated score function when $k = 3$. First, the score function when $k = 3$ for the proposed method can be rewritten as

$$
\begin{aligned}
J &\approx \arg\max_{f^+} \left( \sum_{i=1}^{b} \frac{i}{|S| + 1 - i} \right) U_3(\{S', f^+\}') \\
&= \arg\max_{f^+} \frac{1}{|S| \cdot (|S| - 1)} U_3(\{S', f^+\}'),
\end{aligned}
\tag{22}
$$

where $b = min(|S| + 1 - 3, 3 - 1) = 2$. Equation (22) can be rewritten as

$$
J \approx \arg\max_{f^+} \sum_{f_i \in S} \sum_{f_j \in S} H(f^+, f_i, f_j),
\tag{23}
$$

by the identical process when $k = 2$. Because the newly instantiated score function requires at least two features in $S$, the first and second features are selected based on the score functions when $k = 2$.

Algorithm 2 depicts the incremental search process of the proposed method when $k = 3$. The computational complexity of the proposed method expands to $O(n + n^2 + n^3) = O(n^3)$ due to the $n$, $n^2$, and $n^3$ unit times required for calculating entropy values. With $k = 3$, the proposed method demands additional computational resources to calculate the entropy values, as Equation (23) involves joint entropies among three features. This new score function captures more complex relationships among features by calculating joint entropies of candidate features with all pairs of selected features. However, this results in a significant increase in computational complexity compared to the $k = 2$ method, which has a complexity of $O(n^2)$. Furthermore, estimating the joint entropy between high-dimensional features often requires a large number of patterns to achieve reliable approximations [Lee and Kim, 2015].

