# OpenReview forum: "Unsupervised Feature Selection towards Pattern Discrimination Power"
_auai.org/UAI/2024/Conference — UAI 2024 poster_

### Official Review · Reviewer_Rd5f · 2024-02-27

**Q2-1 Originality-Novelty:** 2
**Q2-2 Correctness-Technical Quality:** 3
**Q2-5 Clarity Of Writing:** 2

**Q1 Summary And Contributions:**

Through an example using a toy dataset, the author demonstrated that contrary to commonly used unsupervised feature selection methods based on minimizing entropy, maximizing entropy could be a new perspective for improving the discriminative power of data patterns. To this end, the author proposed an unsupervised feature selection method based on maximizing entropy. The main innovation in the method is to solve the difficult problem of calculating high-dimensional joint entropy through feature composition when designing the scoring function. Furthermore, extensive experiments demonstrate the performance of the proposed method.

**Q2-3 Extent To Which Claims Are Supported By Evidence:**

3: Good: the main claims are supported by convincing evidence (in the form of adequate experimental evaluation, proofs, (pseudo-)code, references, assumptions).

**Q2-4 Reproducibility:**

4: Excellent: key resources (e.g. proofs, code, data) are available and key details (e.g. proof sketches, experimental setup) are comprehensively described for competent researchers to confidently and easily reproduce the main results.

**Q3 Main Strengths:**

1. The text description in the method part is relatively clear, and the formula description is standardized.
2. The experimental design is relatively complete, and the experimental results prove the effectiveness of the method.

**Q4 Main Weakness:**

1. The innovation point of the method is relatively weak. Although maximizing entropy is a new perspective, feature decomposition and incremental search are both commonly used techniques.
2. The research background is not written clearly enough, the importance of the field of feature selection is not highlighted, and the research motivation for unsupervised feature selection is not given. In the context described by the authors, why not use supervised or semi-supervised feature selection methods that may perform better?
3. The logic of the abstract, introduction and related work parts is not smooth enough, and the language writing needs to be improved.

**Q5 Detailed Comments To The Authors:**

1. The research motivation and main contributions should be more prominent in the abstract and introduction sections.
2. Each symbol in the methods section should be given an explanation, for example the letter d in section 3.2.
3. Formulas in the experimental section should also be numbered.
4. The display of experimental results can be richer.

**Q9 Complying With Reviewing Instructions:**

Yes

---

> ### Author Rebuttal · Authors · 2024-04-09
>
> We are grateful for the constructive comments provided by the reviewer. In earnest response to the reviewer's comments, we have prepared a detailed reply and have made the **_"full revision note"_** available at the following link: https://github.com/anony259106/uai2024/blob/main/revision_note.pdf
>
> Below, we offer answers and explanations that encapsulate the key points addressed in the revision note, providing a concise overview of our responses to the comments.
>
> (Comment 1) The research motivation and main contributions should be more prominent in the abstract and introduction sections.
>
> (Answer 1) We revised the original manuscript to enhance the research motivation and main contributions in the abstract and introduction sections.
>
> First, we emphasize the research motivation in the abstract.
>
> **[Revised Manuscript]**
>
> Abstract:
> ... Although the opposite way, i.e., maximization of the joint entropy, can also lead to novel insights, studies in this direction are rare.
> **Specifically, in the field of information retrieval, selecting features to maximize entropy enables pattern discrimination within a dataset with fewer features, thereby facilitating the construction of an effective taxonomy and significantly improving retrieval efficiency.** In this work, we first demonstrate how two feature subsets, each obtained by minimizing/maximizing the joint entropy, respectively, are different based on a toy dataset. ...
>
> Next, we highlight the main contributions in the introduction section.
>
> **[Revised Manuscript]**
>
> Introduction:
> (End of the introduction section) Finally, we validated the performance of the proposed UFS method using 30 public datasets and confirmed its superiority in terms of pattern discrimination power-related measures. \textcolor{blue}{The main contributions of this work are summarized as follows:
>
>     1. An information-theoretic UFS method is introduced, which maximizes entropy to identify an effective feature subset, thereby significantly enhancing pattern discrimination power within datasets.
>
>     2. A comparative analysis demonstrates our approach using a toy dataset, illustrating the differences between feature subsets obtained through entropy minimization and maximization, and highlighting the enhanced pattern discrimination power achieved with entropy maximization.
>
>     3. To tackle the computational challenge of high-dimensional joint entropy, we introduce a novel score function for UFS based on joint entropy decomposition. This approach effectively decomposes high-dimensional entropy into the sum of lower-dimensional terms, offering a practical approximation for entropy calculation.
>
>     4. The efficacy of the proposed method is validated through extensive testing on 30 public datasets, which confirms its superiority in improving pattern discrimination power over existing UFS methods.
>
> (Comment 2) Each symbol in the methods section should be given an explanation, for example the letter d in section 3.2.
>
> (Answer 2) The letter $d$ in Section 3.2 represents the number of entire features in the dataset.
> We will carefully revise the manuscript to provide a detailed explanation of the symbols.
> For example, $d$ can be clarified as $|F|$ in the revised manuscript.
>
> **[Original Manuscript]**
>
> Let $W \in \mathcal{R}^d$ be the original dataset with $d$ features $F = \{f_1, f_2, \cdots, f_d\}$ and the goal of the UFS is to identify a feature subset $S$ consisting of $n$ features with the optimal pattern discrimination power where $n$ is the number of features to be selected.
>
>
> **[Revised Manuscript]**
>
> Let $W \in \mathcal{R}^{|F|}$ be the original dataset with $d$ features $F = \{f_1, f_2, \cdots, f_{|F|}\}$ and the goal of the UFS is to identify a feature subset $S$ consisting of $n$ features with the optimal pattern discrimination power where $n$ is the number of features to be selected.
>
>
> (Comment 3) Formulas in the experimental section should also be numbered.
>
> (Answer 3) We appreciate the valuable comment to enhance the readability of the manuscript. We will number the formulas in the experimental section.
> An example of the revised manuscript can be found in the **revision note (Page 19)**.
>
>
> (Comment 4) The display of experimental results can be richer.
>
> (Answer 4) We will improve the display of experimental results by adjusting the resolutions of the figures and tables in the revised manuscript.
> We appreciate the valuable comment to enhance the quality of the manuscript.

---

### Official Review · Reviewer_EZKd · 2024-03-06

**Q2-1 Originality-Novelty:** 3
**Q2-2 Correctness-Technical Quality:** 3
**Q2-5 Clarity Of Writing:** 4

**Q10 Ethical Concerns:**

It does not apply.

**Q1 Summary And Contributions:**

The paper deals with unsupervised feature selection by maximizing the joint entropy. The proposed approach is sound and the results are promising.

**Q2-3 Extent To Which Claims Are Supported By Evidence:**

3: Good: the main claims are supported by convincing evidence (in the form of adequate experimental evaluation, proofs, (pseudo-)code, references, assumptions).

**Q2-4 Reproducibility:**

3: Good: key resources (e.g. proofs, code, data) are available and key details (e.g. proofs, experimental setup) are sufficiently well-described for competent researchers to confidently reproduce the main results.

**Q3 Main Strengths:**

The paper is well-written and presents a different approach for feature selection. The experiments consider recent approaches as baselines and the results are promising.

**Q4 Main Weakness:**

The toy example (Table I) is interesting, but it should present some quantitative evaluation. Another point concerns the experimental section. The authors could consider a labeled dataset and treat it as unlabeled. After finding the most representative features, a supervised classification algorithm could be used to compute the accuracy, which is way more insightful than an entropy value.

**Q5 Detailed Comments To The Authors:**

Please, se above.

**Q9 Complying With Reviewing Instructions:**

Yes

---

> ### Author Rebuttal · Authors · 2024-04-09
>
> We are grateful for the constructive comments provided by the reviewer. In earnest response to the reviewer's comments, we have prepared a detailed reply and have made the **_"full revision note"_** available at the following link: https://github.com/anony259106/uai2024/blob/main/revision_note.pdf
>
> Below, we offer answers and explanations that encapsulate the key points addressed in the revision note, providing a concise overview of our responses to the comments.
>
> (Comment 1) The toy example (Table I) is interesting, but it should present some quantitative evaluation.
>
> Answer 1) We describe a quantitative evaluation of the toy example.
> Table 1: Joint entropies of all feature pairs in Table 1 from the original manuscript.
>
> | Selected Features | Entropy | min/max |
> |-------------------|---------|---------|
> | $[f_1, f_2]$        | 1.252   | min     |
> | $[f_1, f_3]$        | 1.918   |         |
> | $[f_1, f_4]$        | 1.459   |         |
> | $[f_2, f_3]$        | 2.252   |         |
> | $[f_2, f_4]$        | 1.459   |         |
> | $[f_3, f_4]$        | 2.585   | max     |
>
> Table 1 presents the joint entropies of all possible feature subsets for $|S|=2$ within the toy dataset from the original manuscript. The selected features $[f_1, f_2]$ exhibit the lowest joint entropy, indicating that this feature pair is the least informative and discriminative. As a result, four distinct patterns from the toy dataset become indistinguishable when selecting these features. Conversely, the selected features $[f_3, f_4]$ reveal the highest joint entropy, and all patterns remain distinguishable when these features are selected.
>
> (Comment 2) Another point concerns the experimental section. The authors could consider a labeled dataset and treat it as unlabeled. After finding the most representative features, a supervised classification algorithm could be used to compute the accuracy, which is way more insightful than an entropy value.
>
> (Answer 2) To evaluate the classification performance of the selected features, we conducted experiments on labeled datasets and attached them in the Appendix section. Detailed results can be found in the **revision note (Pages 14-16)**.
>
> First, we conducted experiments on 30 datasets, using the naive Bayes and decision tree classifiers. Specifically, the classification accuracy is measured by the 10-fold cross-validation with the naive Bayes classifier and the decision tree classifier, which were trained by the data composed of the selected features. In the case of the naive Bayes classifier, the proposed method achieved superior classification accuracy on 14 out of 30 datasets, with an average rank of 1.97. This performance outperformed that of the next most effective method, FMI, which garnered an average rank of 2.57. For the decision tree classifier, the proposed method also led the field, securing the highest classification accuracy on 9 out of the 30 datasets and an average rank of 2.09. This surpassed the second-best method, EGC, which obtained an average rank of 2.27.
>
> Furthermore, we report the classification accuracy achieved by the proposed and compared methods across varying feature sizes on the Alzheimer, Audiology, Chess, and SCADI datasets. As a result, the classification accuracy of the proposed method surpassed that of the compared methods across all four datasets as the feature size increased. This superior performance can be attributed to the feature subset that is optimized towards pattern discrimination power, thereby contributing to improved classification accuracy.

---

### Official Review · Reviewer_aJ19 · 2024-03-22

**Q2-1 Originality-Novelty:** 3
**Q2-2 Correctness-Technical Quality:** 3
**Q2-5 Clarity Of Writing:** 3

**Q1 Summary And Contributions:**

Unsupervised feature selection (UFS) aims to identify crucial features from a dataset based on its inherent properties. Information-based score functions are employed for this purpose, emphasizing essential feature identification. Traditional unsupervised feature selection typically revolves around minimizing the entropy of the chosen feature subset, while exploration of maximizing joint entropy is less common. This study illustrates the disparity between feature subsets achieved through entropy minimization and maximization using a simplified dataset, highlighting the latter's potential for enhancing pattern discrimination. Finally, the authors propose a novel score function, and its pattern discrimination efficacy across various datasets is shown in the experiments.

**Q2-3 Extent To Which Claims Are Supported By Evidence:**

3: Good: the main claims are supported by convincing evidence (in the form of adequate experimental evaluation, proofs, (pseudo-)code, references, assumptions).

**Q2-4 Reproducibility:**

2: Fair: key resources (e.g. proofs, code, data) are unavailable but key details (e.g. proof sketches, experimental setup) are sufficiently well-described for an expert to confidently reproduce the main results.

**Q3 Main Strengths:**

The paper is well-structured and written.

The problem is addressed correctly.

The experimental results show great superiority of the proposed method vs those included in the comparison.

**Q4 Main Weakness:**

The necessity of a discretization method for the features could imply some arbitrariety and also loses of the original information within the data.

The normalization for the other methods and the non-normalization could make the comparison unfair, some discussion and justification should be given about that.

Some discussion on the value for k is missing

**Q5 Detailed Comments To The Authors:**

The phrase "For compared methods, all datasets are normalized to the range of [0, 1] before the FS process, as the authors suggested" doesn't seem supported when reading the original papers for the other methods. They used normalized versions of the metrics sometimes but that doesn't mean that the data are normalized, or at least I don't interpret that. I am concerned that the superiority of the performance is because of this transformation, so i suggest to apply the other methods (EGC, U2, VCSD and FMI) with the original data, non-normalize and see if they behave better.

There is no cpu time reported and not an exhaustive complexity study. I think reporting execution time would be of interest to compare all the alternative techniques.

I would appreciate that the authors indicate if the following references are appropriate to be included:

[ref-1] Wang, Xiaohong, Yidi He, Lizhi Wang, and Zhongxing Wang. "An unsupervised feature selection method based on information entropy." In 2018 3rd International Conference on System Reliability and Safety (ICSRS), pp. 35-39. IEEE, 2018.
 ---> In my opinion this is a very strongly related idea

[ref-2] Zhao, Jie, Jia-ming Liang, Zhen-ning Dong, De-yu Tang, and Zhen Liu. "Accelerating information entropy-based feature selection using rough set theory with classified nested equivalence classes." Pattern Recognition 107 (2020): 107517.

[ref-3] Oreski, Dijana, Stjepan Oreski, and Bozidar Klicek. "Effects of dataset characteristics on the performance of feature selection techniques." Applied Soft Computing 52 (2017): 109-119.

--> this one is about classification (supervised) but some ideas could be translated to the current topic

 Some questions:

 - When you say :'In our experiments, we set k to two because it is the minimum value for the score function being a multivariate feature filter 2', I would like to know the implications (in terms of the quality of the results and on complexity) of using values of k larger than 2.

 - When you decide to use this discretization: 'equal-width binning method [Talukdar et al.,2018] where the number of bins is set to the number of classes for each numerical dataset.', two issues are implicitly raised:

 	1.- why this discretization method? why not equal-frequency? Why don't you use a (distinct) rule to select the number of bins? This decision can change the results, so proper justification has to be given
 	2.- aren't you somehow transforming the problem into supervised? I mean, in the moment the number of bins for the discretization to be applied to the numerical variables is decided by the number of classes, we are not using an Unsupervised approach anymore.

 - Could you provide some insight on using values for k in Equation (7) bigger than 2?

AFTER AUTHORS' REPLY: I have read the point-to-point comments and the results for the related experiments, and my main concerns have been appropriately addressed. I  appreciate the effort and that all the raised questions have been considered and answered. I have raised my overall score.

**Q9 Complying With Reviewing Instructions:**

Yes

---

> ### Author Rebuttal · Authors · 2024-04-09
>
> We are grateful for the constructive comments provided by the reviewer. In earnest response to the reviewer's comments, we have prepared a detailed reply and have made the **_"full revision note"_** available at the following link: https://github.com/anony259106/uai2024/blob/main/revision_note.pdf
>
> Below, we offer answers and explanations that encapsulate the key points addressed in the revision note, providing a concise overview of our responses to the comments.
>
> (Comment 1) I am concerned that the superiority of the performance is because of this transformation, so i suggest to apply the other methods (EGC, U2, VCSD and FMI) with the original data, non-normalize and see if they behave better.
>
> (Answer 1) We conducted additional experiments on the five numerical datasets (Alzheimer, Lsvt, Umist, WarpAR10P, WarpPIE10p) to evaluate the performance of the compared methods without normalization. Furthermore, as in Comment 5, all datasets are discretized using the equal-width binning method with 10 bins for the proposed method. Results can be found in the **revision note (Page 9)**.
> The proposed method consistently outperforms the other methods in terms of entropy and SDP across all datasets, even without normalization. Nevertheless, we acknowledge the reviewer's comments for the rigorous evaluation of the proposed method and will include the results in the revised manuscript.
>
> (Comment 2) There is no cpu time reported and not an exhaustive complexity study. I think reporting execution time would be of interest to compare all the alternative techniques.
>
> (Answer 2) We present a comparative analysis of the execution times for conducted UFS methods in the **revision note (Page 10)**. In the same experimental conditions, the proposed method ranks as the second-fastest method on average after EGC, underscoring its efficiency in a wide range of datasets. Also, for algorithms such as EGC, which rely on specific parameter settings, the time to achieve optimal results can increase multiple times depending on the complexity of the parameter space being navigated.
>
> (Comment 3) I would appreciate that the authors indicate if the following references are appropriate to be included:
>
> (Answer 3) We appreciate the relevant references to enhance the quality of the manuscript, and we are likely to include the references in the revised manuscript.
>
> (Comment 4) I would like to know the implications (in terms of the quality of the results and on complexity) of using values of k larger than 2.
>
> (Answer 4) We provide a detailed explanation of the implications of using values of $k$ larger than 2 by giving a concrete example by introducing a newly instantiated score function when $k=3$ in the **revision note (Page 11)**.
> The newly instantiated score function when $k=3$ is defined as follows:
> \begin{equation}
>   J \approx argmax_{f^+} \sum_{f_i \in S} \sum_{f_j \in S} H(f^+, f_i, f_j).
> \end{equation}
>
> by the identical process in the original manuscript. Furthermore, we provide an incremental search process for the proposed method when $k=3$ in the **revision note (Page 12)**. The computational complexity of the proposed method expands to $O(n+n^2+n^3) = O(n^3)$ because $n, n^2, n^3$ unit times are consumed for calculating entropy values.
> The new score function can capture more complex relationships among features because it calculates joint entropies of candidate features with all pairs of selected features. However, the computational complexity increases significantly compared to the original proposed method with $k=2$, which is $O(n^2)$. Also, an estimation of the joint entropy between high-dimensional features often requires a large number of patterns to achieve reliable approximations.
>
> (Comment 5) When you decide to use this discretization: 'equal-width binning method where the number of bins is set to the number of classes for each numerical dataset.',
> 1. why this discretization method? why not equal-frequency? Why don't you use a (distinct) rule to select the number of bins?
> 2. aren't you somehow transforming the problem into supervised?
>
> (Answer 5) The reason we avoid the equal-frequency binning method is due to its effect of allocating an equal number of patterns to each bin, which results in a uniformly distributed transformation. Therefore, when selecting the first feature during the incremental search, scores for all features become identical which can degrade the quality of $S$.
> Also, we appreciate the valuable comment on the discretization method. We will adjust the discretization method to ensure that the proposed method remains a UFS method in the revised manuscript. Currently, we report the results of the proposed method with the equal-width binning method with 10 bins in Answer 1.
>
> (Comment 6) Could you provide some insight on using values for k in Equation (7) bigger than 2?
>
> (Answer 6) We provided a detailed explanation of the implications of using values of $k$ bigger than 2 in Answer 4.

---

### Official Review · Reviewer_GMsc · 2024-03-26

**Q2-1 Originality-Novelty:** 2
**Q2-2 Correctness-Technical Quality:** 2
**Q2-5 Clarity Of Writing:** 2

**Q1 Summary And Contributions:**

the paper proposes to select a feature set by maximizing the entropy. k-cardinality set approximation is proposed, and empirical results show better performance, in term of entropy and SDP, than existing baselines.

**Q2-3 Extent To Which Claims Are Supported By Evidence:**

2: Fair: the main claims are somewhat supported by evidence (but the experimental evaluation may be weak, or does not match entirely with the claims, important baselines may be missing, proofs contain important ideas but lack rigor, algorithmic details are only discussed superficially, references are imprecise, assumptions are not sufficiently motivated or explicated, etc.).

**Q2-4 Reproducibility:**

2: Fair: key resources (e.g. proofs, code, data) are unavailable but key details (e.g. proof sketches, experimental setup) are sufficiently well-described for an expert to confidently reproduce the main results.

**Q3 Main Strengths:**

- contain theoretical results on approximations
- better results than baselines (although not sure the objective here)

**Q4 Main Weakness:**

- lack the clarify on the objective and evaluation metric
- downstream tasks should be used in experiments, such as cluster (as discussed in introduction) or another relevant task.

**Q5 Detailed Comments To The Authors:**

- " ...search for a feature subset by optimizing": What is the motivation for maximizing the entropy? Only explanation is found later.
- "the original feature set F should have the largest entropy": how does the large entropy leads to more discriminative power?
- It is not clear what objective of unsupervised FS is. Is it a subset that preserve the most information, instead of discriminative power?
- the theoretical proof seem to be borrowed from existing papers, without explanation. It is not clear if anything new here. At best, they are not self contained.
- Eq 9 to 12: are they the same approximation or further approximations from each equation above?
- Experiment evaluation metric: authors did not clarify the objective, hence i'm not sure what would maximizing SDP and entropy would help in any downstream task. What are some applications? Authors should also test the selected features' performance on these applications.
- What is SDP and entropy for the entire feature set? In addition, it is not clear if the baseline methods also seek to maximize the entropy or have different objectives.

**Q9 Complying With Reviewing Instructions:**

Yes

---

> ### Author Rebuttal · Authors · 2024-04-09
>
> We are grateful for the constructive comments provided by the reviewer. In earnest response to the reviewer's comments, we have prepared a detailed reply and have made the **_"full revision note"_** available at the following link: https://github.com/anony259106/uai2024/blob/main/revision_note.pdf
>
> Below, we offer answers and explanations that encapsulate the key points addressed in the revision note, providing a concise overview of our responses to the comments.
>
> (Comment 1) "...search for a feature subset by optimizing": What is the motivation for maximizing the entropy?
>
> (Answer 1) The main goal of maximizing entropy is to use the distinctiveness of patterns to improve information retrieval and taxonomy construction. The proposed method selects features with the highest entropy value based on information theory, enhancing the pattern discrimination power. Constructing the taxonomic tree and distributing patterns based on prioritized features with high entropy leads to a reduction in the tree's maximum depth and ensures an even distribution of patterns throughout the tree, making data analysis more straightforward.
>
> (Comment 2) how does the large entropy leads to more discriminative power?
>
> (Answer 2) We provide a detailed theoretical justification, which can be found on the **revision note (Pages 2-5)**.
> Our proof begins by revising the SDP formulation in relation to the dataset $W$ and the feature subset $S$, illustrating the effect of pattern distinctiveness on SDP. Through specific examples, we demonstrate how varying numbers of indistinct patterns influence $H(S)$, establishing a correlation between SDP and entropy: a decrease in SDP, indicative of an increase in the number of indistinct patterns, implies a reduction in entropy.
>
> (Comment 3) Is it a subset that preserve the most information, instead of discriminative power?
>
> (Answer 3) We provide theoretical proof of the relationship between the entropy of a feature subset and its SDP, showing that a decrease in SDP implies a decrease in entropy, as elaborated in Answer 2.
>
> (Comment 4) The theoretical proof seem to be borrowed from existing papers, without explanation.
>
> (Answer 4) Equation (6) in the manuscript presents that the upper bound of $k$-cardinality entropy is determined by the $k-1$-cardinality entropy of $S'$, thereby the high-dimensional joint entropy $U_k(S')$ can be decomposed into the lower-dimensional joint entropy by applying Equation (6) recursively.
> Equation (7) represents the general form of the above recursive decomposition that can directly calculate the upper bound of $H(S)$ with $k$-cardinality. While these referenced papers focus on minimizing the entropy of the feature subset to reduce feature redundancy, our approach aims to maximize the entropy of the feature subset, thereby amplifying the pattern discrimination power.
> We would like to add this explanation to the revised manuscript to enhance the clarity of the theoretical proof.
>
> (Comment 5) Eq 9 to 12: are they the same approximation or further approximations from each equation above?
>
> (Answer 5) We appreciate the reviewer's comment to clarify the approximation process of the score function. Equation (10) represents a further approximation of Equation (9) by omitting the coefficient and constant term.  Equation (11) retains the same level of approximation as Equation (10) since $U(f^+, S')$ encompasses the sum of the joint entropies between $f^+$ and every feature in $S'$. Equation (12) parallels the approximation in Equation (11), as when $S=\emptyset$, it results in the selection of only the feature with the maximum entropy.
>
> (Comment 6) What are some applications? Authors should also test the selected features' performance on these applications.
>
> Maximizing $H(S)$ facilitates the construction of an efficient taxonomic tree by minimizing the need for discriminators. Our method iteratively selects features to maximize the entropy of the subset, aiming for a uniform distribution across the tree, which results in a structure resembling a balanced tree. A detailed example in the **revision note (Pages 6-8)** demonstrates how our approach yields a more balanced tree compared to methods that minimize entropy.
>
> (Comment 7) What is SDP and entropy for the entire feature set? In addition, it is not clear if the baseline methods also seek to maximize the entropy or have different objectives.
>
> (Answer 7) Both the SDP and entropy exhibit monotonic increases with the inclusion of additional features. We provide a detailed explanation in the **revision note (Page 8)** that the SDP and entropy of the entire feature set are maximized when all features are selected.
> EGC, U2, and VCSD methods select features based on manifold learning, taking into account the weight of each feature in the learning process. Meanwhile, FMI shares the same objective of using entropy maximization as our proposed method, yet its score function is derived from a heuristic method.

---

### Meta-Review · Area_Chair_wYKg · 2024-04-09

To solve the difficult problem of calculating high-dimensional joint entropy through feature composition when designing the scoring function，this paper proposes an unsupervised feature selection method based on maximizing entropy. The experimental results validate the effectiveness of the proposed method. This paper received four reviews with "Strong accept", "Accept", "Weak Accept" and "Borderline reject". The averaged overall score achieves 6.25. I Initiated and monitored discussions. The authors provided their rebuttals to the main weakness from the reviewers and the only reviewer who gives "Borderline reject" has not responsed to the authors' Rebuttals till now.